# Systematic identification of molecular mediators of interspecies sensing in a community of two frequently coinfecting bacterial pathogens

Tiffany M. Zarrella [1,2], Anupama Khare [1] *

**1** Laboratory of Molecular Biology, Center for Cancer Research, National Cancer Institute, National Institutes of Health, Bethesda, Maryland, United States of America, **2** Postdoctoral Research Associate Training Program, National Institute of General Medical Sciences, National Institutes of Health, Bethesda, Maryland, United States of America

* anupama.khare@nih.gov

**Data Availability Statement:** The RNA-seq data are available from the NCBI Gene Expression Omnibus (GEO) (https://www.ncbi.nlm.nih.gov/

## Abstract

Bacteria typically exist in dynamic, multispecies communities where polymicrobial interactions influence fitness. Elucidating the molecular mechanisms underlying these interactions is critical for understanding and modulating bacterial behavior in natural environments. While bacterial responses to foreign species are frequently characterized at the molecular and phenotypic level, the exogenous molecules that elicit these responses are understudied. Here, we outline a systematic strategy based on transcriptomics combined with genetic and biochemical screens of promoter-reporters to identify the molecules from one species that are sensed by another. We utilized this method to study interactions between the pathogens *Pseudomonas aeruginosa* and *Staphylococcus aureus* that are frequently found in coinfections. We discovered that *P. aeruginosa* senses diverse staphylococcal exoproducts including the metallophore staphylopine (StP), intermediate metabolites citrate and acetoin, and multiple molecules that modulate its iron starvation response. We observed that StP inhibits biofilm formation and that *P. aeruginosa* can utilize citrate and acetoin for growth, revealing that these interactions have both antagonistic and beneficial effects. Due to the unbiased nature of our approach, we also identified on a genome scale the genes in *S. aureus* that affect production of each sensed exoproduct, providing possible targets to modify multispecies community dynamics. Further, a combination of these identified *S. aureus* products recapitulated a majority of the transcriptional response of *P. aeruginosa* to *S. aureus* supernatant, validating our screening strategy. Cystic fibrosis (CF) clinical isolates of both *S. aureus* and *P. aeruginosa* also showed varying degrees of induction or responses, respectively, which suggests that these interactions are widespread among pathogenic strains. Our screening approach thus identified multiple *S. aureus* secreted molecules that are sensed by *P. aeruginosa* and affect its physiology, demonstrating the efficacy of this approach, and yielding new insight into the molecular basis of interactions between these two species.

geo/) under accession numbers GSE185963 and GSE186138.

**Funding:** This work was supported by the Intramural Research Program of the NIH, National Cancer Institute, Center for Cancer Research. TMZ was supported by a Postdoctoral Research Associate Training (PRAT) Fellowship award 1FI2GM137843-01 from the National Institute of General Medical Sciences. The funders had no role in study design, data collection and analysis, decision to publish, or preparation of the manuscript.

**Competing interests:** The authors have declared that no competing interests exist.

**Abbreviations:** BIP, 2,2′-bipyridyl; CAS, chromeazurol S; CF, cystic fibrosis; cpm, cycles per minute; DFX, deferoxamine; DTPA, diethylene triamine penta-acetic acid; GEO, Gene Expression Omnibus; GO, Gene Ontology; IS, isotopic standard; NTML, Nebraska Transposon Mutant Library; PsP, pseudopaline; RNA-seq, RNA sequencing; StP, staphylopine; TCA, tricarboxylic acid; WT, wild type.

## Introduction

Bacteria frequently exist in multispecies communities in which cooperative and competitive interactions govern community composition and physiological outputs. Interactions among pathogenic bacterial species in multispecies infections can affect disease outcomes and antibiotic susceptibility [1,2]. Elucidating the molecules underlying these interactions is therefore critical to understand and modulate bacterial behavior in communities, and pairwise interaction studies using unbiased genetic and biochemical approaches are powerful tools for such analyses [3–5].

Previous studies on pairwise interspecies interactions have focused on the identification of molecules produced by one species and the effect of these molecules on specific phenotypes of another species, such as growth, biofilm formation, or antibiotic resistance [6–12]. Recent work has described the genome-wide response of one species to another at the molecular level using either transcriptomics, proteomics, or metabolomics in multispecies conditions, revealing that these responses can be complex and affect numerous pathways [13–15], and are therefore likely mediated by multiple sensed secreted factors.

One powerful strategy to determine the cues being sensed by bacteria is to infer them from the cellular response. Examples include the SOS response indicating the sensing of DNA damage or the heat shock response revealing the perception of high environmental temperature [16,17]. In multispecies communities, specific bacterial responses have been used to identify the interspecific mediators. These include the up-regulation of oxidative stress pathways in response to secreted redox-active molecules, alleviation of nutritional requirements due to sensing and use of secreted amino acids or other nutrients, and induction of iron starvation pathways by sensed iron competition caused by siderophores secreted by a foreign species [18–20]. However, most such studies of interactions typically focus on a single response and the underlying molecules.

The goal of this work was to develop a comprehensive approach to define which exoproducts from one species are sensed by another and use this framework to study interspecies sensing in a model two-species system consisting of *Pseudomonas aeruginosa* and *Staphylococcus aureus*. Both are opportunistic pathogens that coexist in wound infections, nosocomial pneumonia, as well as lung infections in cystic fibrosis (CF) patients [21–24], where coinfection with both pathogens has been associated with increased disease severity [25–27]. Identifying the molecular determinants of interactions between these two coinfecting pathogens can thus provide novel insights into persistent infections and multispecies communities in general.

Interactions between these two species have been extensively studied (reviewed by [28–31]). *P. aeruginosa* affects the growth, metabolism, antibiotic resistance, and transcriptional state of *S. aureus*, mainly via the secretion of multiple antimicrobials [6,32–35]. *S. aureus* induces exploratory motility in *P. aeruginosa* via unknown exoproducts, and the quorum sensing molecule autoinducer-2 and cell wall precursor *N*-acetyl glucosamine that can be produced by *S. aureus* affect the regulation of *P. aeruginosa* virulence factors [36–42]. However, characterization of the *P. aeruginosa* transcriptional response to *S. aureus* and the comprehensive identification of the underlying sensed *S. aureus* secreted factors and their production pathways have not been fully accomplished.

In this study, we outline a strategy to systematically identify the secreted molecules from one species that are sensed by another, utilizing transcriptomics in conjunction with genetic and biochemical screens of promoter-reporters. Our approach revealed that *P. aeruginosa* senses at least four distinct signals secreted by *S. aureus*: the metallophore staphylopine (StP), which induces zinc starvation, the intermediate metabolites citrate and acetoin, which induce their uptake and catabolism, and unidentified molecules that affect iron-related pathways.

Through the genome-wide screen, we also delineate the *S. aureus* genes that alter the production of these sensed molecules. Further, we find that these staphylococcal exoproducts mediate both antagonistic and beneficial effects on *P. aeruginosa* physiology. StP participates in metal competition and abrogates *P. aeruginosa* biofilm formation, while the intermediate metabolites can serve as sole carbon sources, and the iron-chelating secreted factors deliver iron to *P. aeruginosa*. Several clinical isolates of *S. aureus* and *P. aeruginosa* show induction and responses, respectively, of varying intensity, and other bacterial species can also induce some of the same pathways in *P. aeruginosa*, indicating the generality of these phenomena. Finally, we show that these *S. aureus* secreted molecules explain a major part of the *P. aeruginosa* response to *S. aureus*, demonstrating the utility of our approach to identify the secreted factors that underlie interspecies interactions.

## Results

### Framework to comprehensively define the molecular mediators of bacterial interspecies sensing

We developed a systematic methodology within a two-species system to identify the sensed foreign molecules that underlie the global response of one species to the other (Fig 1A). Our approach first defines the complex response resulting from exposure to a foreign species and identifies the different individual components of the response. Then, these individual responses are used to identify the respective causal foreign molecules via unbiased genetic and biochemical screens. Unlike previous approaches, our strategy does not focus on the effect of individual molecules on a foreign species, or on globally identifying all molecules that are produced by a species, rather each response is used to identify the respective specific foreign molecule(s) that together make up the interspecies interaction.

In the initial step, global transcriptional analysis is performed on one species exposed to another species or its cell-free supernatant compared to monocultures, to identify which pathways are differentially regulated by interspecies sensing. Next, promoter-reporter constructs are designed using representative genes from these up-regulated classes. These promoter-reporter strains are then employed in two complementary approaches to identify the sensed interspecific molecules. In the first approach, an arrayed transposon mutant library in the species being sensed is screened to determine the mutants that are deficient in inducing reporter expression. The mutants are likely to be involved in the regulation, biosynthesis, or secretion of the cues, and mutant gene function or characterization of the mutant supernatant is therefore used to identify the sensed molecules. In the second, the sensed supernatant is fractionated to identify fractions that contain the active molecules inducing the promoters that can then be further analyzed by additional fractionation and mass spectrometry. This two-pronged unbiased approach has the potential to comprehensively reveal the molecules that lead to complex responses in a foreign species, irrespective of which pathways and mechanisms constitute the response. We applied this scheme to study the *S. aureus* secreted products that are sensed by *P. aeruginosa*.

### *P. aeruginosa* senses *S. aureus* secreted products and up-regulates metal- and metabolite-related pathways

We focused on the response of *P. aeruginosa* to *S. aureus* cell-free supernatant, as opposed to a coculture model, since *P. aeruginosa* produces antimicrobials that may kill and lyse *S. aureus* cells [29,49], thus affecting the relative levels of staphylococcal secreted products that may be present. To determine the response of *P. aeruginosa* PA14 to *S. aureus* JE2, we performed

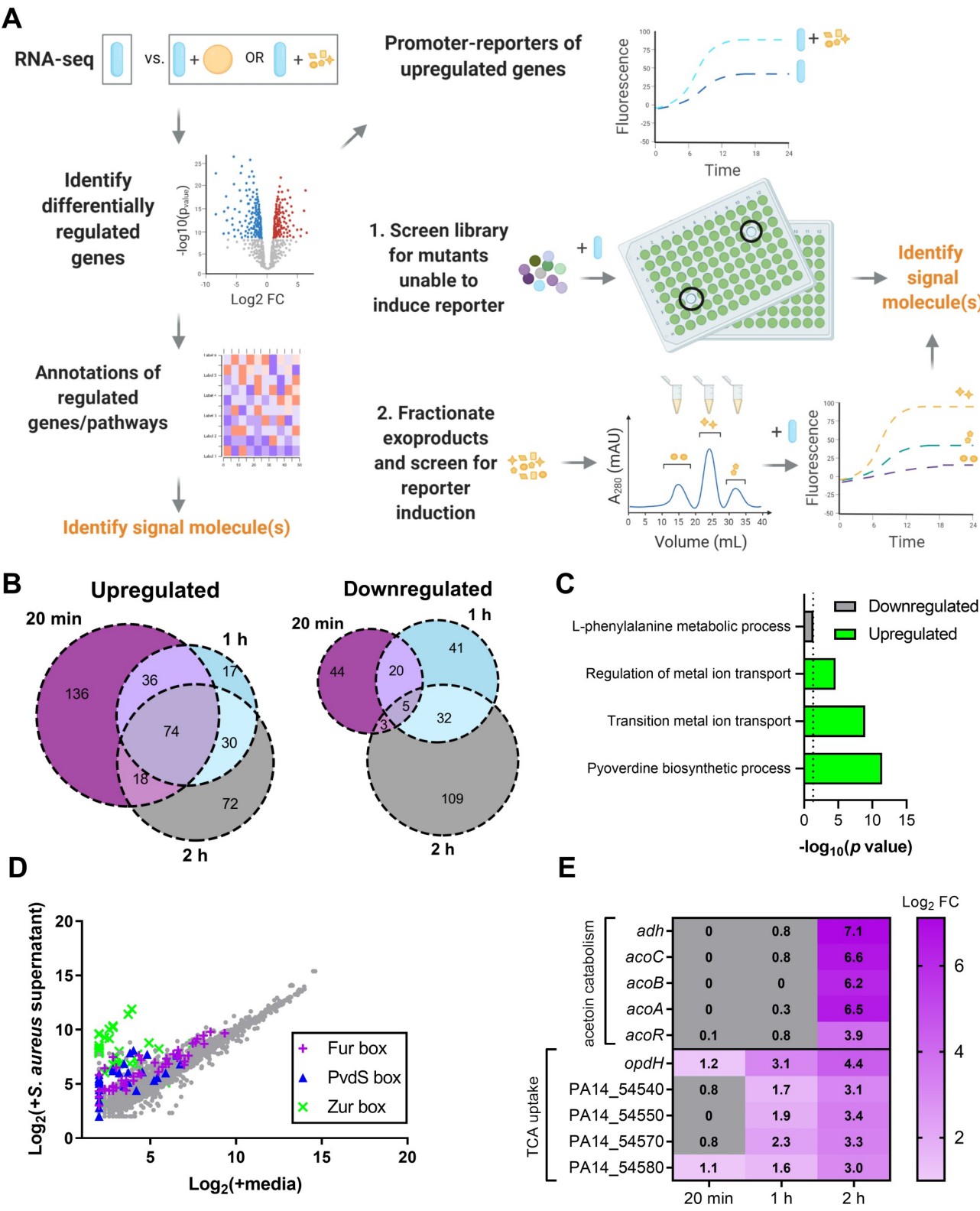

**Fig 1. *P. aeruginosa* differentially regulates metal deprivation and intermediate metabolite uptake pathways in the presence of *S. aureus* supernatant.** (**A**) Schematic for identification of molecular mediators of interspecies sensing in a two species system. Global transcriptional response of one species to another species, or its secreted exoproducts, compared to monocultures, is determined by RNA-seq. Analysis of the differentially regulated genes and pathways may be used to identify the signaling molecules that are sensed. Promoter-reporters are constructed from representative

up-regulated genes and used to screen for the signaling molecules by 2 complementary methods. In the first method, a mutant library is screened for mutants that disrupt production or export of the signaling molecules and therefore have lower reporter induction. In the second, the supernatant is biochemically fractionated and fractions are screened for induction of the promoter. **(B)** Venn diagrams of up-regulated and down-regulated genes (Log$_2$ fold change $\geq$ 1 or $\leq$ −1 and $p$ < 0.05 cutoff) in *P. aeruginosa* after *S. aureus* supernatant exposure compared to media control after 20 minutes, 1 hour, and 2 hours. **(C)** GO enrichment of differentially expressed genes in *P. aeruginosa* after 20 minutes [43–45]. Nonredundant categories for down-regulated and up-regulated biological processes are shown. **(D)** Scatterplot of mean expression levels of transcripts after *S. aureus* supernatant exposure compared to media control after 20 minutes. Genes annotated previously in *P. aeruginosa* strain PAO1 as being regulated by Fur, PvdS (IS box), or Zur are shown [46–48]. **(E)** Log$_2$ fold change of select transcripts in metabolite-uptake operons that increase in abundance over time. The data underlying panels B, C, D, and E can be found in Tables B, C, and D, and both Tables A and B, in S1 File, respectively. GO, Gene Ontology; RNA-seq, RNA sequencing.

RNA sequencing (RNA-seq) on early-log phase *P. aeruginosa* cultures at 20 minutes, 1 hour, and 2 hours after the addition of 25% (v/v) *S. aureus* spent media or the same volume of media as a control. We found approximately 100 to 200 genes that were significantly differentially expressed between these 2 conditions at each time point (**Fig 1B** and **Tables A and B** in **S1 File**).

We identified the major pathways that are induced in *P. aeruginosa* upon sensing *S. aureus* molecules to define candidate promoters for the screening strategy (**Fig 1A**). First, we performed Gene Ontology (GO) enrichment analysis of the differentially regulated genes in *P. aeruginosa* after *S. aureus* exposure [43–45]. The down-regulated genes showed enrichment of amino acid metabolism genes while up-regulated genes indicated significant enrichment in genes involved in metal ion transport, as well as the biosynthesis of the siderophore pyoverdine, suggesting the sensing of metal deprivation (**Fig 1C** and **Table C** in **S1 File**). Several genes previously reported to be controlled by the Zn-dependent repressor Zur, the Fe-dependent repressor Fur, and the Fe-specific sigma factor PvdS in *P. aeruginosa* PAO1 were up-regulated 20 minutes after *S. aureus* supernatant exposure (**Fig 1D** and **Table D** in **S1 File**) [46–48]. Next, to identify additional robustly up-regulated pathways that may not be well described by GO terms, we focused on all the up-regulated genes that increased in fold change over time upon *S. aureus* supernatant exposure. Of the 461 total up-regulated genes, 73 increased over time across the 3 time points (**S1 Fig**). We selected the operons that showed the highest terminal fold change, and these were associated with intermediate metabolite pathways, namely acetoin catabolism and tricarboxylic acids (TCA) uptake (**Figs 1E and S1**).

The strongest *P. aeruginosa* response to *S. aureus* cell-free supernatant was thus represented by 4 pathways: Zn deprivation, Fe deprivation, TCA uptake, and acetoin catabolism. We selected one promoter from each of these four pathways whose respective transcripts had distinct patterns of expression in the RNA-seq analysis (PA14_11320, *pvdG*, *opdH*, and *acoR*, respectively) (**Fig 2A**) and designed promoter-reporters using the fluorescent protein mScarlet. Each reporter led to significantly higher levels of mScarlet expression in the presence of *S. aureus* supernatant compared to the media control and displayed dose-dependent responses (**Fig 2B**). The two metal-deprivation promoter-reporter strains induced the reporter earlier in the supernatant than the control, while the two intermediate metabolite reporter strains induced the reporter almost exclusively upon *S. aureus* supernatant exposure.

Addition of 25% (v/v) *S. aureus* supernatant did not change the pH of the medium (**S2A Fig**), and *P. aeruginosa* growth did not change due to addition of 25% (v/v) *S. aureus* supernatant in the first 6 hours when promoter-reporter induction was measured (**S2B Fig**), suggesting that the *P. aeruginosa* response to *S. aureus* was not due to pH or growth differences. Further, the promoter-reporters were not induced by 25% (v/v) media salts lacking nutrients, indicating that this is unlikely to be a general starvation response (**S2C Fig**). Conversely, to test if addition of nutrients would disrupt induction, *S. aureus* supernatant was lyophilized and resuspended in media or water and added at 25% (v/v) to the promoter-reporter strains. All 4

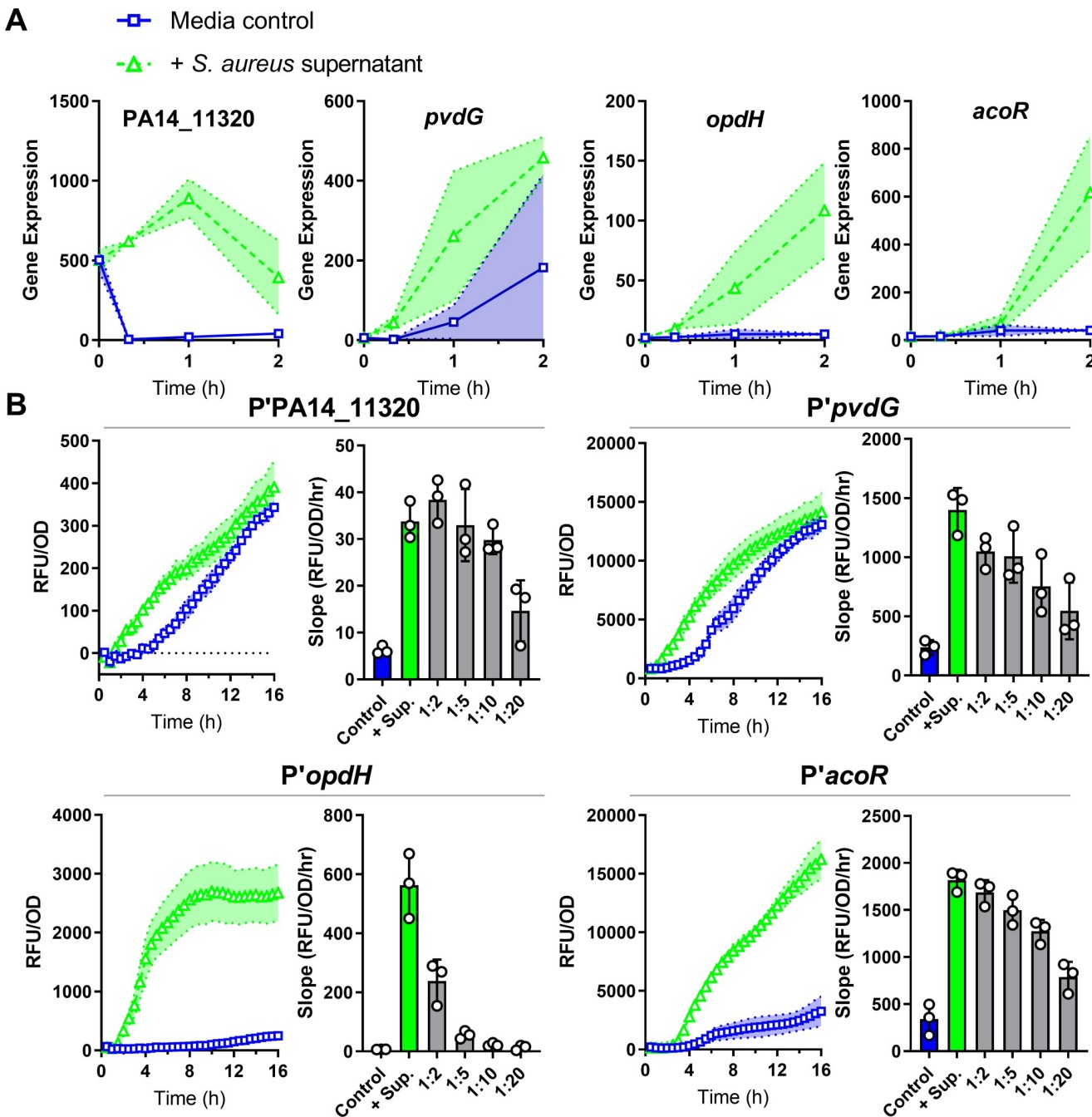

**Fig 2. *P. aeruginosa* promoter-reporters are induced by *S. aureus* exoproducts. (A)** Expression levels from *P. aeruginosa* RNA-seq after exposure to *S. aureus* supernatant or media control for four transcripts as gene candidates for promoter-reporter construction. Gene annotations: PA14_11320, Zur-regulon putative lipoprotein in operon with ABC transporter; *pvdG*, pyoverdine biosynthesis thioesterase; *opdH*, TCA uptake porin; *acoR*, transcriptional regulator of acetoin catabolism genes. **(B)** RFUs of mScarlet normalized to OD$_{600}$ over time after exposure to *S. aureus* supernatant or media control in the indicated promoter-reporter strains. Slope calculated from 1.5 to 5 hours (promoters of PA14_11320, *opdH*, and *acoR*) or 1 to 4 hours (promoter of *pvdG*) after addition of supernatant, supernatant dilutions, or media control. Each strain had higher expression of mScarlet after supernatant addition compared with media control and displayed dose-dependent responses. Shaded regions and error bars show the SD. The data underlying panels A and B can be found in Table A in S1 File and S1 Table, respectively. RFU, relative fluorescence unit; RNA-seq, RNA sequencing; SD, standard deviation.

promoter-reporter strains were still induced (**S2C Fig**), although induction of the *pvdG* promoter brought on by iron starvation was lower when lyophilized exoproducts were resuspended in media, likely due to increased iron availability. In addition, induction of the *acoR* promoter was lower in response to lyophilized supernatant, regardless of resuspension in media or water, suggesting that lyophilization may reduce active metabolite concentrations.

## The metallophore activity of *S. aureus* StP up-regulates the Zn-deprivation response in *P. aeruginosa*

Each promoter-reporter strain was used to screen supernatants from the Nebraska Transposon Mutant Library (NTML), which contains over 1,800 ordered transposon mutants in the methicillin-resistant *S. aureus* strain JE2 [50]. We reasoned that transposon insertions in the genes involved in the expression, biosynthesis, or export of the sensed product(s) would result in altered promoter induction.

Using the promoter of a putative lipoprotein under control of the Zur regulon (PA14_11320) [48] to screen the NTML for mutants that were deficient in inducing mScarlet reporter expression, we found that 2 of the top hits (after excluding mutants with growth defects) were mutants deficient for the biosynthesis or export of the multimetal binding molecule StP (**Table E in S1 File**). We therefore tested mutants for the entire pathway and found that mutants with transposon insertions in the genes encoding either the StP exporter (*cntE*) or biosynthetic enzymes (*cntKLM*) were deficient in induction of the promoter-reporter (**Fig 3A and 3B**). StP is a nicotianamine-like molecule that is exported out of the cell, where it binds zinc and imports it back into the cell [51]. StP is generated from $_L$-histidine in 3 stepwise reactions by histidine racemase CntK, nicotianamine synthase CntL, and opine dehydrogenase CntM which incorporates pyruvate at the terminal step [51,52]. StP supports growth in zinc-deficient conditions, and mutants with transposon insertions in the StP biosynthesis or export genes showed growth defects in the presence of the zinc chelator *N,N,N′,N′*-tetrakis(2-pyridinylmethyl)-1,2-ethanediamine (TPEN) (**S3A Fig**) [53,54].

*P. aeruginosa* produces a similar secreted opine metallophore, pseudopaline (PsP), which is also important for zinc uptake, and is produced by CntL and CntM with the addition of a terminal -ketoglutarate instead of pyruvate (**Fig 3A**) [52,55–57]. PsP is exported by CntI in the inner membrane and MexAB-OprM in the outer membrane and is imported back into the cell by CntO [56,58]. The entire *cntOLMI* operon was up-regulated upon exposure to *S. aureus* supernatant (**Tables A and B in S1 File**). Along with P′PA14_11320, we examined StP-mediated induction of 2 other Zur-regulated promoters: *cntO* (PA14_63960), the first gene in the *cnt* operon, and *dksA2* (PA14_73020), which encodes a transcription factor in an operon with cobalamin synthesis genes [59]. *S. aureus cnt* mutants lacking StP reduced mScarlet reporter expression from these 3 Zur-regulated promoters compared to wild type (WT) (**Fig 3B**). The addition of increasing amounts of StP to the *S. aureus cntM*::tn mutant supernatant restored P′PA14_11320 induction to WT levels at concentrations greater than 40 μM (**Fig 3C**), indicating that this concentration is sufficient to induce the promoter to a level similar to that of WT supernatant. Next, we sought to characterize if StP chelates metals from or delivers metals to *P. aeruginosa*. Addition of the zinc chelator TPEN increased promoter induction, with 0.5 μM added to *S. aureus cntM*::tn mutant supernatant mimicking the induction seen upon addition of WT *S. aureus* supernatant (**Figs 3D and S3B**). Inversely, the induction was decreased by the addition of zinc, cobalt, or nickel (**Figs 3E and S3C**), suggesting that StP chelates these metals, thereby inducing the zinc starvation response in *P. aeruginosa*.

To examine the role of the similar *P. aeruginosa* opine metallophore PsP in the response to StP, induction of P′PA14_11320 was measured in *P. aeruginosa* mutants lacking the genes

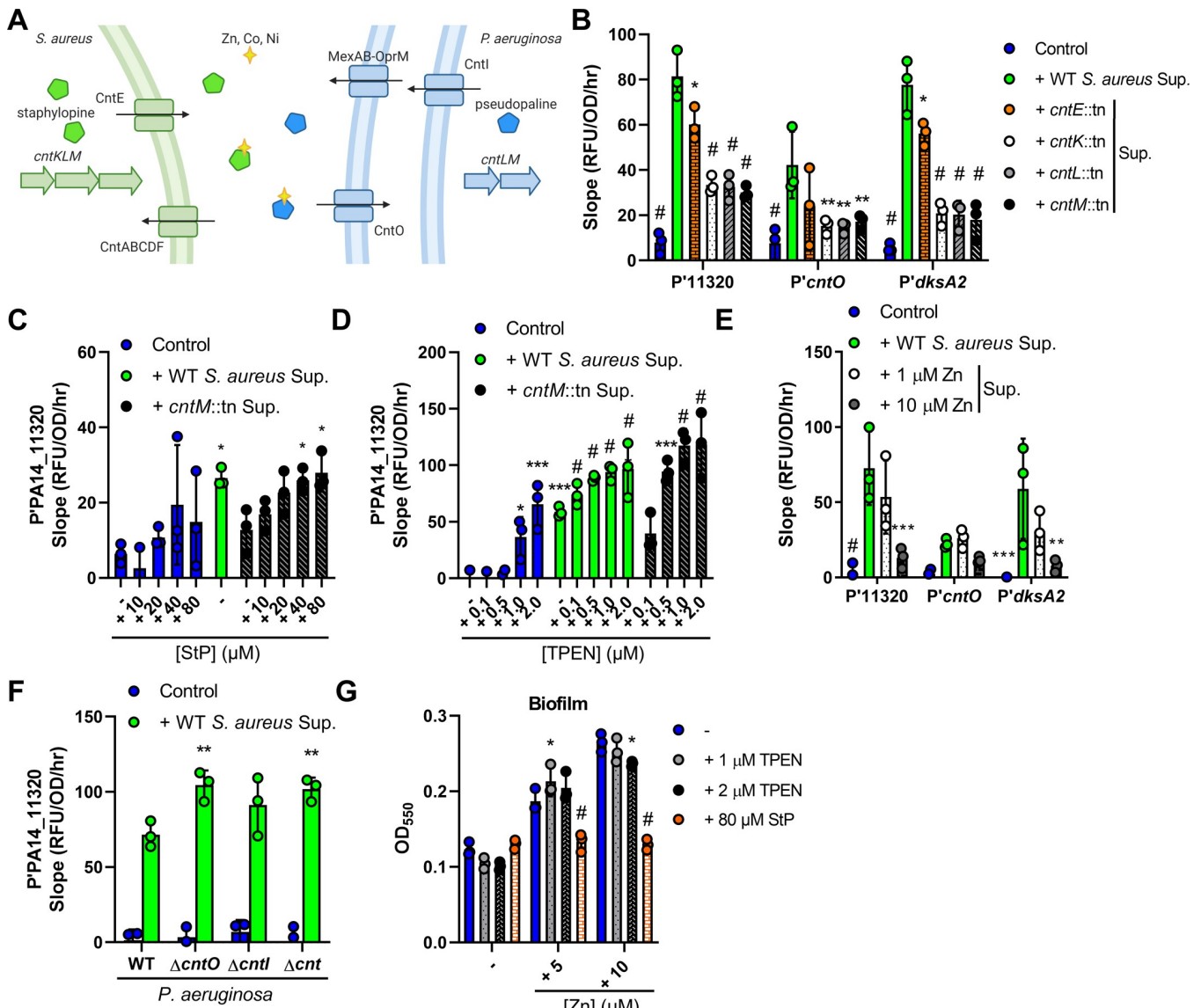

**Fig 3. The *S. aureus* metallophore StP is sensed by *P. aeruginosa*. (A)** Screening of the NTML identified mutants in the biosynthesis and secretion of StP as deficient in the induction of the PA14_11320 promoter. Schematic shows StP (green pentagon) and PsP (blue pentagon) biosynthesis, export, binding to transition metals Zn, Co, Ni, and uptake. **(B–F)** RFU of mScarlet normalized to OD$_{600}$ over time from 1.5 to 5 hours after exposure to media control or *S. aureus* WT or *cnt* supernatant and/or addition of the indicated concentrations of StP, Zn, or TPEN in the indicated *P. aeruginosa* WT or mutant promoter-reporter strains (promoters of PA14_11320, *cntO*, or *dksA2*). **(G)** *P. aeruginosa* cultures were grown in medium alone or with the additives listed, and biofilm formation was assessed by crystal violet staining after 24 hours of growth. Data shown for all panels are the means of 3 independent biological replicates. Datasets were analyzed by 2-way ANOVA with Tukey test for multiple comparisons (significance shown for comparisons to: B, WT supernatant; F, respective WT sample, respectively) or 1-way ANOVA with Dunnett test for multiple comparisons to the respective control (E, WT supernatant; C, D, and G, respective media control). The error bars denote the SD. *, $p < 0.05$; **, $p < 0.01$; ***, $p < 0.001$; #, $p < 0.0001$. The data underlying panels B–G can be found in S1 Table. NTML, Nebraska Transposon Mutant Library; PsP, pseudopaline; RFU, relative fluorescence unit; SD, standard deviation; StP, staphylopine; WT, wild type.

encoding the importer (*cntO*), exporter (*cntI*), or entire *cnt* operon for PsP. Upon supernatant addition, induction of mScarlet reporter expression was increased in the *P. aeruginosa* Δ*cntO* and Δ*cnt* mutants compared to the WT (**Fig 3F**), demonstrating that the presence of PsP partially protects from StP-induced zinc starvation, indicating that the 2 metallophores compete for zinc. Thus, in response to the opine metallophore StP, *P. aeruginosa* induces the Zur regulon that includes genes to synthesize its own zinc metallophore, PsP.

Higher levels of zinc availability are associated with increased *P. aeruginosa* biofilm formation and increased antagonism of *S. aureus* [60–62]. We therefore explored the effects of StP, which we expect leads to zinc starvation, on *P. aeruginosa* biofilm formation and *P. aeruginosa*–*S. aureus* polymicrobial interactions. *P. aeruginosa* was grown in static conditions in medium with 0, 5, or 10 μM zinc added and zinc addition increased biofilm formation (**Fig 3G**). While the addition of 1 or 2 μM TPEN had only modest effects on biofilm formation, 80 μM StP reversed the effects of zinc addition and inhibited biofilm formation (**Fig 3G**). A major component of *P. aeruginosa*–*S. aureus* interactions are *P. aeruginosa*–produced antimicrobials that inhibit *S. aureus* growth [29]. To assess the role of StP on the inhibition of *S. aureus* growth by *P. aeruginosa*, cell-free supernatants were collected from *P. aeruginosa* cultures grown in media alone or with the addition of 10 μM zinc, 1 μM TPEN, or 80 μM StP, and the growth of *S. aureus* in 50% (v/v) supernatant was determined. While supernatant from *P. aeruginosa* grown in zinc or TPEN inhibited *S. aureus* survival similar to the control supernatant, supernatant from *P. aeruginosa* cultured with StP showed lower inhibition of *S. aureus*, suggesting that StP may affect *P. aeruginosa* antimicrobial production (**S3D Fig**).

## Staphylococcal secreted products affect siderophore-biosynthesis responses in *P. aeruginosa*

To determine the secreted products affecting the iron starvation response, the *S. aureus* transposon library was screened with the promoter of a pyoverdine synthesis gene, *pvdG*, which is induced by low-iron conditions [47]. Since the *P. aeruginosa* Zur regulon was responding to a metallophore, we hypothesized that the *pvdG* promoter would similarly respond to a siderophore, but none of the *S. aureus* siderophore mutants reduced induction (**Table F in S1 File**). Instead, we identified more than 100 mutants each that had a higher (up-regulating) or lower (down-regulating) induction of the reporter compared to the WT (**Fig 4A and Table F in S1 File**), and the relative iron chelation of the mutant supernatants did not correlate with their promoter induction ($r = −0.04$) (**Figs 4B and S4A and Table G in S1 File**). Several down-regulating supernatants had significantly higher chelation than WT, including those from mutants of the ATP-dependent proteases ClpC and ClpP, suggesting that these proteases contribute to the regulation of chelating factor(s) (**Fig 4B and Table G in S1 File**).

It has been reported that *P. aeruginosa* can utilize specific iron-chelating xenosiderophores such as deferoxamine (DFX) to obtain iron [63–66], and P′*pvdG* was repressed by DFX (**S4B Fig**), showing iron delivery to *P. aeruginosa*. As a control, another iron chelator diethylene triamine penta-acetic acid (DTPA), which chelates iron away from *P. aeruginosa*, induced P′*pvdG* (**S4B Fig**). Thus, the lack of correlation between P′*pvdG* induction and relative chelation by the *S. aureus* mutant supernatants may be due to a combination of iron competition and delivery to *P. aeruginosa*.

Next, we employed the second approach in our strategy to determine the sensed secreted products (**Fig 1A**) where we biochemically fractionated *S. aureus* supernatant by size-exclusion chromatography and then screened concentrated fractions for their ability to induce the P′*pvdG* promoter-reporter. We also treated each fraction with proteinase K to determine if the active molecule(s) contain peptide bonds. Relative to the water control, fraction 4 increased promoter induction, but this was abolished by proteinase K treatment, while fraction 5 only induced the promoter after proteinase K treatment (**Fig 4C**). Additionally, fractions 6 to 12 decreased induction regardless of proteinase K treatment (**Fig 4C**). Similar to our observations with the mutant supernatants, fraction 4 did not have any chelating activity, whereas the fractions that decreased promoter induction (fractions 6 to 12) did (**Fig 4D**). This suggests that the

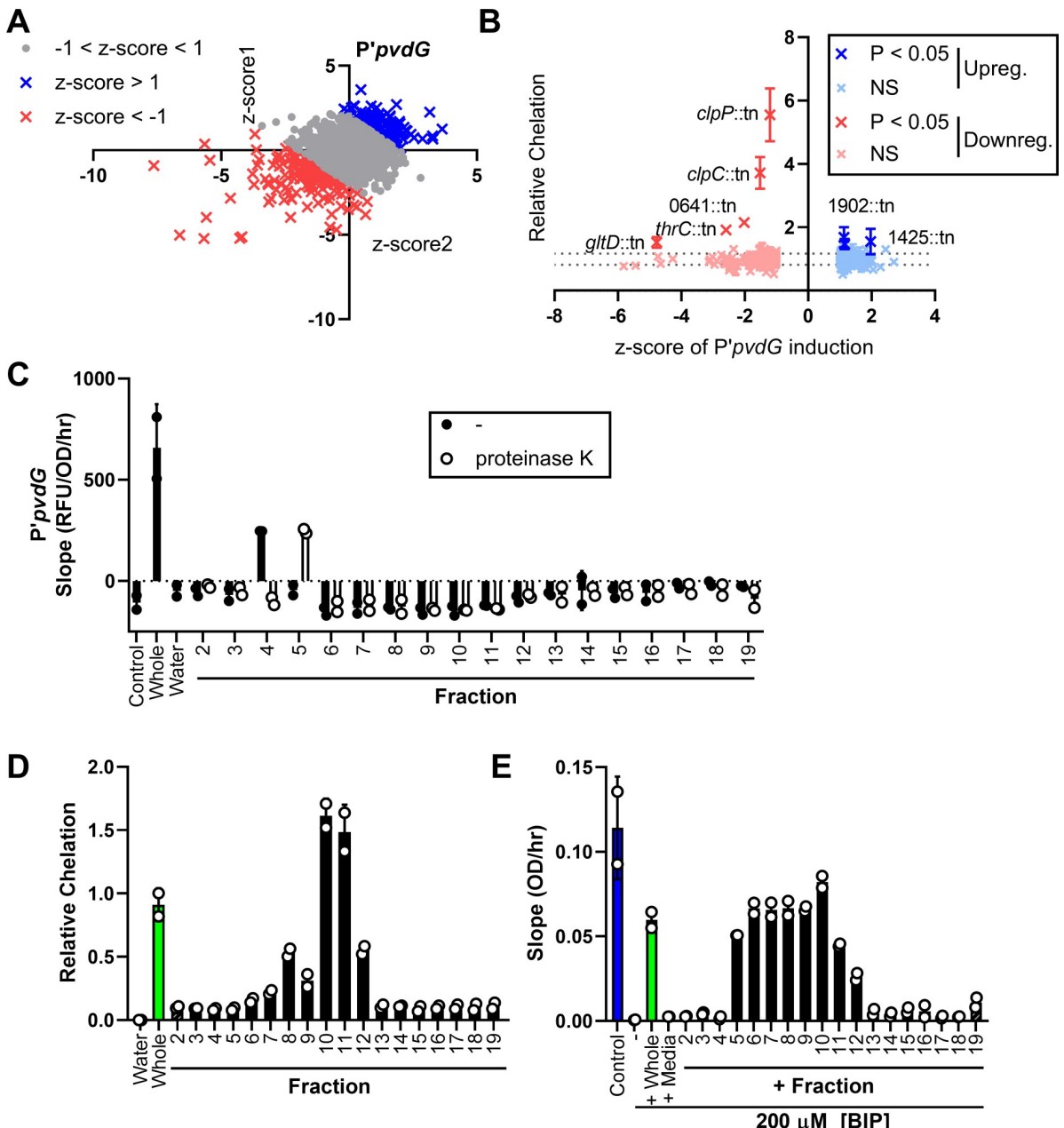

**Fig 4. *P. aeruginosa* pyoverdine promoter induction is affected by multiple *S. aureus* secreted factors. (A)** Transposon library screen with P'*pvdG*-reporter strain to determine mutations in *S. aureus* that increase or decrease induction in *P. aeruginosa*. For each mutant supernatant, the z-scores calculated from the mean slope (RFU/OD$_{600}$/hr) of each respective experiment for 2 replicates of the screen are shown. (z-score < −1, down-regulating, red; z-score > 1, up-regulating, blue). **(B)** Relative chelation of the down-regulating and up-regulating supernatants ($r = −0.04$ for correlation between the relative chelation and the z-score). Data were analyzed by 1-way ANOVA with Dunnett test for multiple comparisons to the WT control. Supernatants with significantly different chelation compared to the WT control are labeled. Data shown represent 3 independent replicates. **(C–E)** WT *S. aureus* supernatant was fractionated by size-exclusion chromatography. Two independent biological replicates are shown. **(C)** RFU of P'*pvdG*-mScarlet normalized to OD$_{600}$ over time from 1 to 4 hours after exposure to media control, 1X whole *S. aureus* WT supernatant, water, or 10X fractions with or without proteinase K treatment (mixed 1:1 with medium). **(D)** Relative chelation of water control, whole supernatant, or 10X fractions. **(E)** Slope of OD$_{600}$ over time from 1 to 8 hours of the *P. aeruginosa* Δ*pvdJ* Δ*pchE* strain in medium control or medium with 200 μM BIP with the addition of whole WT *S. aureus* supernatant, media, or 10X fractions. Error bars denote the SD for panels B–E. The data underlying panels A and B can be found in Tables F and G in S1 File, respectively, and data for panels C–E can be found in S1 Table. RFU, relative fluorescence unit; SD, standard deviation; WT, wild type.

chelators present in the down-regulating fractions repressed the expression of pyoverdine bio-synthesis genes, possibly due to delivery of iron to *P. aeruginosa*.

To test if secreted staphylococcal molecules could deliver iron to *P. aeruginosa*, we used a *P. aeruginosa* Δ*pvdJ* Δ*pchE* mutant, which cannot produce the siderophores pyoverdine and pyoche-lin. The Δ*pvdJ* Δ*pchE* mutant is unable to grow in medium containing 200 μM of the iron chelator 2,2′-bipyridyl (BIP). The addition of DFX, but not DTPA, supports the growth of this mutant in BIP media (**S4C Fig**), confirming the ability of *P. aeruginosa* to utilize DFX to take up iron in the absence of its own siderophores. Interestingly, the Δ*pvdJ* Δ*pchE* mutant grew in BIP media after addition of whole *S. aureus* supernatant or individual fractions 5 to 12, but not media (**Figs 4E** and **S4C**). Together with the promoter-reporter and iron chelation assays, these results indicate that one or more *S. aureus* chelators deliver iron to *P. aeruginosa*, while at least one staphylococcal exoproduct without chelating activity induces pyoverdine production.

## *S. aureus* secreted intermediate metabolites citric acid and acetoin induce uptake and catabolism in *P. aeruginosa*

To determine the staphylococcal secreted products that induce expression of the TCA uptake operon, the NTML transposon collection was screened with the *opdH* promoter-reporter strain. OpdH is a porin that uptakes TCAs and is induced by the TCAs isocitrate, cis-aconitate, and citrate [67]. Similar to the screen with the *pvdG* promoter, we obtained more than 100 mutants that up-regulated or down-regulated the *opdH* promoter (**Fig 5A** and **Table H in S1 File**). GO enrichment analysis of these mutants indicated that aspartate family amino acid bio-synthesis genes were enriched in the up-regulating mutants and branched-chain amino acid biosynthesis, metabolism, and cellular respiration pathways were enriched in the down-regu-lating mutants (**Fig 5B** and **Table I in S1 File**) [43–45]. Many of the identified genes encode proteins that catalyze enzymatic reactions within the TCA cycle and related pathways (**Fig 5C** and **Tables H and I in S1 File**). This included the down-regulating supernatant from *gltA*::tn that encodes citrate synthase II, which converts TCA intermediates to citrate, and the up-regu-lating supernatant from *acnA*::tn encoding aconitate hydrase, which breaks down citrate (**Fig 5A** and **Table H in S1 File**), suggesting that *S. aureus*–secreted citrate may induce the *opdH* promoter-reporter. Exogenously added citrate induced the P'*opdH* reporter in a dose-depen-dent manner with between 100 and 200 μM having similar induction to *S. aureus* supernatant (**Fig 5D**), and direct measurements showed that *S. aureus* supernatant contained 166.0 ± 13.0 μM citrate (**Fig 5E** and **Table J in S1 File**). Citrate concentrations in the up-regu-lating and down-regulating mutant supernatants significantly correlated with induction of the P'*opdH* promoter (Pearson's $r = 0.6663$, $p < 0.0001$) (**Fig 5E** and **Table J in S1 File**). Therefore, *S. aureus* secretes the TCA intermediate metabolite citrate, which is sensed by *P. aeruginosa*.

Recently, it was described that acetoin is released by *S. aureus* and in response, *P. aeruginosa* up-regulates acetoin and butanoate pathways, including the *acoR* gene, encoding a transcrip-tional regulator of the acetoin catabolism operon [68–70], which was up-regulated over time after addition of *S. aureus* supernatant. We predicted that a transposon mutant screen of P'*acoR* induction may identify mutations in staphylococcal acetoin production and/or secre-tion. We identified nearly 200 mutants each that induced or repressed the promoter (**Fig 6A** and **Table K in S1 File**); however, no significant pathways were identified by GO enrichment analysis. Two of the most down-regulating mutant supernatants had transposon insertions in *ilvB* and *ilvN*, which encode enzymes that convert pyruvate into α-acetolactate, leading into the butanoate cycle where acetoin and 2,3-butanediol are synthesized and catabolized (**Fig 6A** and **6C** and **Table K in S1 File**). The promoter of *acoR* is responsive to acetoin, 2,3-butane-diol, and α-acetolactate, but not citrate (**Figs 6B** and **S5A**); however, 2,3-butanediol and α-

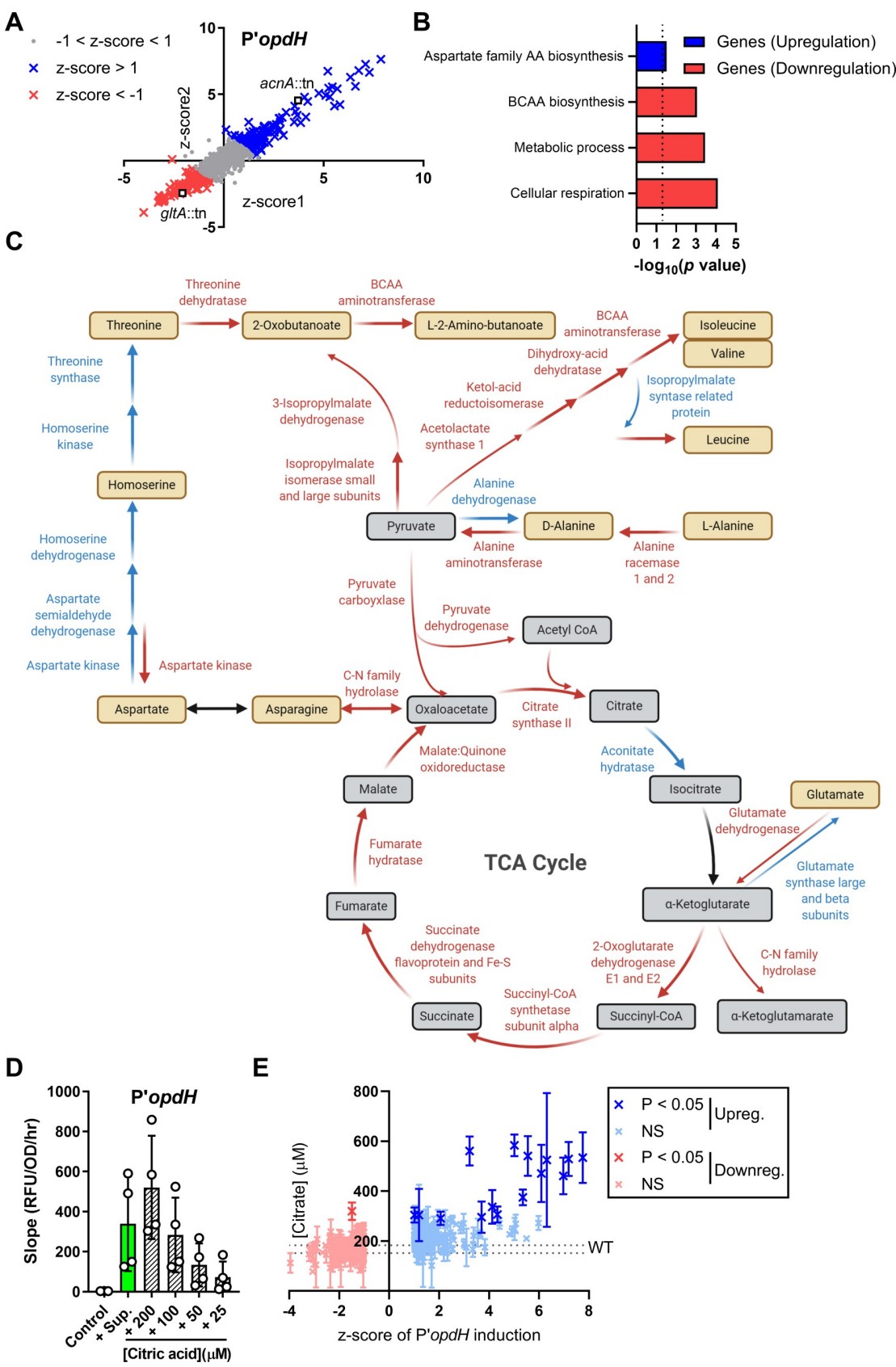

**Fig 5. *S. aureus*–secreted citrate is sensed by *P. aeruginosa*. (A)** Transposon library screen with P′*opdH*-reporter strain to determine mutations in *S. aureus* that increase or decrease induction in *P. aeruginosa*. For each mutant supernatant, the z-scores calculated from the mean slope (RFU/OD$_{600}$/hr) of each respective experiment for two replicates of the screen are shown (z-score < −1, down-regulating, red; z-score > 1, up-regulating, blue). **(B)** GO enrichment analysis of down-regulating and up-regulating mutants. **(C)** Enzymatic reactions in central metabolism are shown, with the colors representing the effect of the mutant supernatants on promoter induction (down-regulating, red; up-regulating, blue; mutant not in library and/or no change, black). **(D)** RFU of P′*opdH*-mScarlet normalized to OD$_{600}$ over time from 1.5 to 5 hours after exposure to media control, *S. aureus* supernatant, or dilutions of citric acid. **(E)** Citric acid measurements of the down-regulating and up-regulating supernatants correlate to their induction of the P′*opdH* promoter (r = 0.6663). Data were analyzed by 1-way ANOVA with Dunnett test for multiple comparisons to the WT control. Data shown represent 3 independent replicates for panels DE. Error bars denote the SD for panels DE. The data underlying panels AC, B, D, and E can be found in Table H in S1 File, Table I in S1 File, S1 Table, and Table J in S1 File, respectively. GO, Gene Ontology; RFU, relative fluorescence unit; SD, standard deviation; TCA, tricarboxylic acid; WT, wild type.

acetolactate could not be detected in *S. aureus* supernatant in quantities that would be needed to induce the promoter (**S5B Fig**). The acetoin concentration in the regulating mutant supernatants correlated with promoter induction in most supernatants (Pearson's r = 0.2157, p < 0.0001), including those of genes leading into the butanoate cycle (**Fig 6C and 6D**). However, a few select down-regulating mutant supernatants contained significantly higher acetoin levels than WT, such as *sucD* encoding succinyl-CoA synthase and the *clpC* and *clpP* mutants (**Fig 6D** and Table L in **S1 File**). Therefore, the main staphylococcal metabolite that induces *acoR* expression is acetoin.

To ascertain if these intermediate metabolites can be utilized as carbon sources by *P. aeruginosa*, 1 mM citrate, 2,3-butanediol, and/or acetoin, or 1 mM glucose as a positive control were added to the growth medium as the sole carbon source and *P. aeruginosa* growth was monitored for 16 hours. As expected, in the absence of any carbon sources, *P. aeruginosa* was unable to grow (k = 0.003), while it grew with the addition of glucose (k = 1.057) (**Fig 6E**). *P. aeruginosa* was also able to grow after addition of citric acid, 2-3-butanediol, and acetoin in combination (k = 0.781), or with just citric acid (k = 0.625), while slight growth was seen after the addition of 2,3-butanediol or acetoin alone (k = 0.273 and 0.270) (**Fig 6E**). Thus, *P. aeruginosa* can utilize these secreted intermediate metabolites as energy sources.

## Clinical isolates of *S. aureus* and *P. aeruginosa* also show identified induction and response phenomena

Through the promoter-reporter screens, we identified four distinct pathways in *P. aeruginosa* that are induced by *S. aureus* JE2 secreted products. To test how widespread the production of these molecules is, we surveyed supernatants from four clinical isolates of *S. aureus* that were monoisolated (CF049 strain) or coisolated with *P. aeruginosa* (CF061, CF085, and CF089 strains) from 3 different CF patients. Supernatants from these strains induced all 4 promoter-reporters in *P. aeruginosa* (**Fig 7A**), albeit to different degrees. While all the strains contained chelators, only CF049 and CF061 supported siderophore-deficient *P. aeruginosa* growth in iron-restricted medium (**Fig 7B and 7C**). CF089, which showed the highest relative chelation, was unable to support the growth of the *P. aeruginosa* siderophore mutant, indicating that *S. aureus* likely produces multiple chelators, only some of which can deliver iron to *P. aeruginosa*, and the relative amounts of these may vary between strains. In addition, citrate and acetoin concentrations in the supernatants largely correlated with induction of the *opdH* and *acoR* promoters, respectively, except for CF061 and CF085 having lower levels of acetoin (**Fig 7D**). Additionally, to test how widespread the identified *P. aeruginosa* responses are, four *P. aeruginosa* clinical isolates (CF17, CF33, CF72, and CF104) that were coisolated with *S. aureus* from four different CF patients were transformed with each promoter-reporter construct. All strains induced all reporters after exposure, with the exception of the *acoR* and *pvdG* promoter-

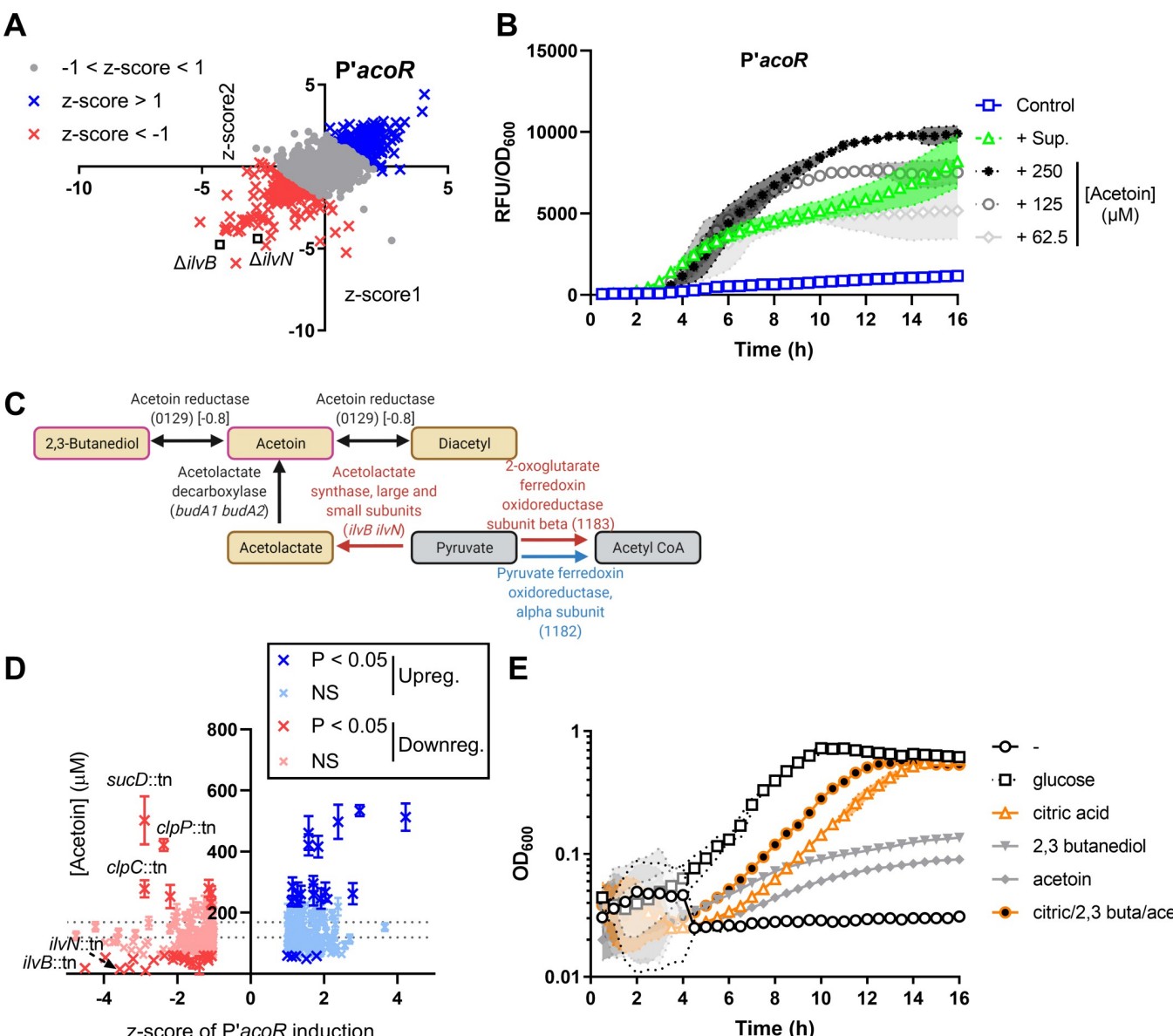

**Fig 6. Intermediate metabolites produced by *S. aureus* are sensed and utilized as carbon sources by *P. aeruginosa*. (A)** Transposon library screen with P′*acoR*-reporter strain to determine mutations in *S. aureus* that increase or decrease induction in *P. aeruginosa*. For each mutant supernatant, the z-scores calculated from the mean slope (RFU/OD$_{600}$/hr) of each respective experiment for 2 replicates of the screen are shown (z-score < −1, down-regulating, red; z-score > 1, up-regulating, blue). **(B)** RFU per OD$_{600}$ of P′*acoR*-mScarlet after addition of media, *S. aureus* supernatant, or the indicated concentrations of acetoin. **(C)** Enzymatic reactions of butanoate metabolism are shown, with the colors representing the effect of the mutant supernatants on promoter induction (down-regulating, red; up-regulating, blue; mutant not in library and/or no change, black). **(D)** Acetoin measurements of the up-regulating and down-regulating supernatants correlate with P′*acoR* induction ($r = 0.2157$). Acetoin concentration data were analyzed by 1-way ANOVA with Dunnett test for multiple comparisons to the WT control. For panels B and D, data shown represent 3 independent replicates and error bars denote the SD. **(E)** Growth of *P. aeruginosa* was monitored at OD$_{600}$ after the addition of 1 mM glucose, citric acid, 2,3-butanediol, acetoin, or a mix of citric acid, 2,3-butanediol, and acetoin as the sole carbon sources. Two independent replicates are shown. Shaded regions denote the SD. The data underlying panels AC and D can be found in Tables K and L in S1 File, and data for panels B and E can be found in S1 Table. RFU, relative fluorescence unit; SD, standard deviation; WT, wild type.

reporters in CF72 (**S6 Fig**). These data indicate that the molecular mechanisms we identified from our approach are prevalent in *S. aureus* and *P. aeruginosa* clinical isolates.

Since the sensing pathways in *P. aeruginosa* were induced by different strains of *S. aureus*, we wondered if other bacterial species could also induce similar responses in *P. aeruginosa*.

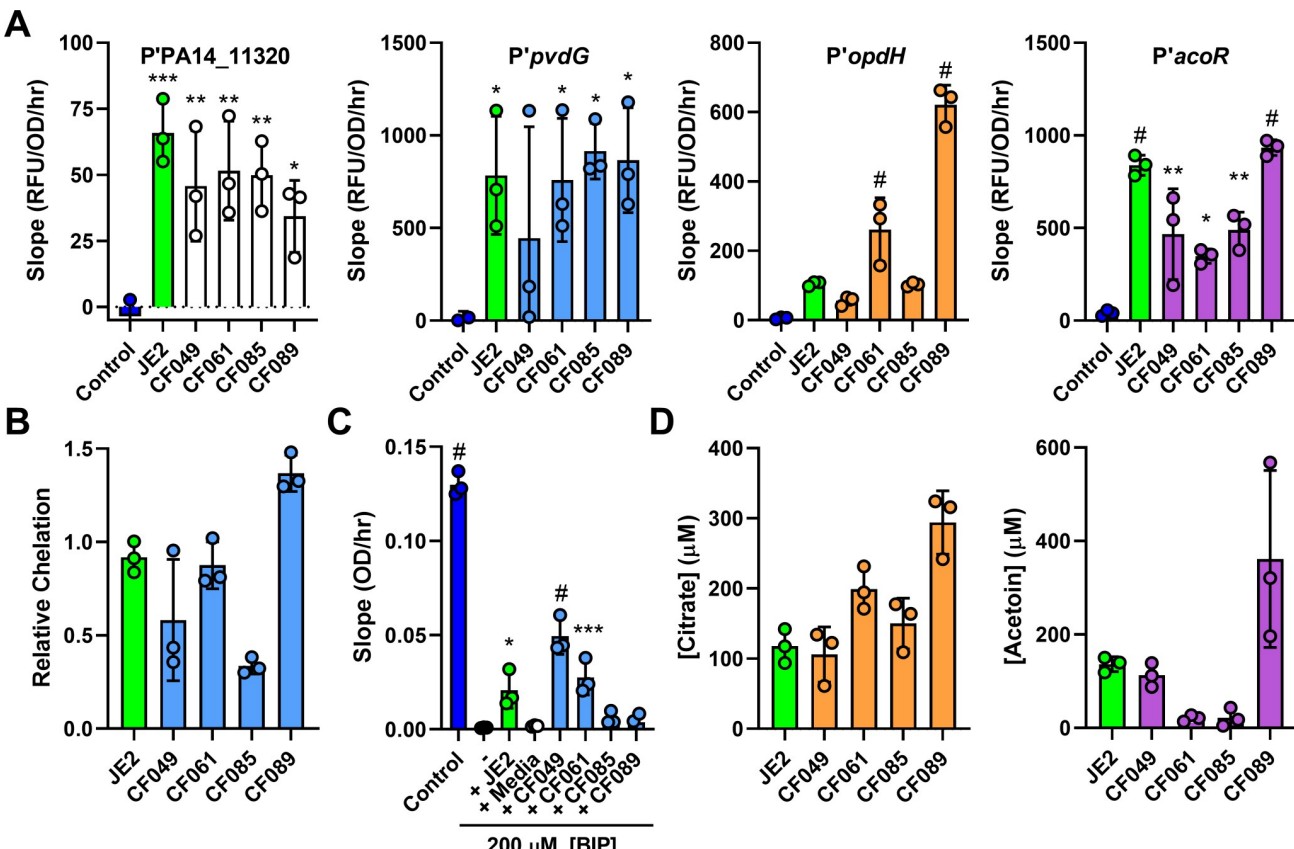

**Fig 7. Sensed exoproducts are secreted by *S. aureus* clinical isolates.** (A) RFU of mScarlet normalized to $OD_{600}$ over time (promoter of PA14_11320, *opdH*, and *acoR* calculated from 1.5 to 5 hours; promoter of *pvdG* calculated from 1 to 4 hours) of the indicated promoter-reporter strains after exposure to media control, or supernatants from *S. aureus* JE2 or clinical isolates CF049, CF061, CF085, and CF089. (B) Relative chelation of supernatants collected from *S. aureus* JE2 or the indicated clinical isolate. (C) Slope of $OD_{600}$ over time from 1 to 8 hours of *P. aeruginosa* Δ*pvdJ* Δ*pchE* strain in medium control or medium with 200 μM BIP with the addition of medium or supernatant from *S. aureus* JE2 or the indicated clinical isolate. (D) Citrate and acetoin measurements of supernatants collected from *S. aureus* JE2 or the indicated clinical isolates. For panels A and C, datasets were analyzed by 1-way ANOVA with Dunnett test for multiple comparisons to the media only control for A and no addition for C. Three independent replicates shown for all panels. Error bars denote the SD. *, $p < 0.05$; **, $p < 0.01$; ***, $p < 0.001$; #, $p < 0.0001$. The data underlying all panels can be found in S1 Table. RFU, relative fluorescence unit; SD, standard deviation.

Therefore, we tested cell-free supernatants from 8 additional species: *Bacillus subtilis*, *Staphylococcus epidermidis*, *Burkholderia cenocepacia*, *Escherichia coli*, *Klebsiella pneumoniae*, *Salmonella enterica* Typhimurium, *Stenotrophomonas maltophilia*, and *Vibrio cholerae*. Induction of the *P. aeruginosa* response pathways varied among the species, with different combinations of 2 to 4 species inducing each pathway (S7 Fig). Notably, *S. epidermidis*, which is most closely related to *S. aureus*, did not induce zinc deprivation, pyoverdine production, or *opdH* in *P. aeruginosa*, whereas *B. cenocepacia* induced all 4 promoters (S7 Fig).

## Identified secreted factors underlie a major part of the *P. aeruginosa* response to staphylococcal exoproducts

To determine if the identified molecules recapitulated the effect of *S. aureus* whole supernatant on *P. aeruginosa*, we added the individual molecules or a combination of all molecules in medium to *P. aeruginosa* cells and performed RNA-seq after 20 minutes and 2 hours incubation (**Table M in S1 File**). The molecules were added at concentrations measured from the supernatant or that induced equivalent promoter activity: 80 μM StP, 150 μM citrate, and

150 μM acetoin. Since we did not identify the molecule(s) that induce the promoter of *pvdG*, we added a mixture of the iron chelators DTPA and DFX at concentrations that represented the induction of mScarlet reporter production (**S4B Fig**). The mixture of the 2 chelators was added to mimic the dual phenotypes we observed, where DTPA induces the *pvdG* promoter, and DFX delivers iron to *P. aeruginosa*. As expected, each of the 4 products induced the respective response pathway in *P. aeruginosa* (**Fig 8A**). Compared to the medium control, addition of the supernatant, the respectively paired identified products, or all products in combination led to up-regulation of the metal starvation genes PA14_11320 and *pvdG* at 20 minutes and intermediate metabolite uptake and metabolism genes *opdH* and *acoR* at 2 hours (**Fig 8A and Table M in S1 File**). DTPA/DFX also induced PA14_11320, potentially due to chelation of zinc by DTPA (**Fig 8A**) [71]. We next determined what proportion of the *P. aeruginosa* response to *S. aureus* supernatant was explained by the identified products. Of the 417 genes that were up-regulated upon exposure to *S. aureus* supernatant, 235 (56.3%) were also up-regulated by at least one of the identified products, while 218 (52.3%) were up-regulated upon exposure to the combination of all products (**Fig 8B**). Despite the sensed products being identified using up-regulated promoter-reporters, of the 263 genes that were down-regulated upon exposure to *S. aureus* supernatant, 133 (50.6%) were also down-regulated by at least one of the identified products, while the combination of all molecules down-regulated 77 genes (29.3%) (**S8A Fig**). Thus, the 4 classes of identified sensed molecules account for a substantial part of both the up-regulated and down-regulated transcriptional responses of *P. aeruginosa* to *S. aureus* supernatant.

Given that we selected our promoter-reporters based on significantly enriched pathways and genes with high fold change of gene expression (**Fig 1C and 1E**), we posited that our identified sensed molecules likely underlie the up-regulation of genes with higher fold change. In fact, the genes up-regulated by both *S. aureus* supernatant and at least one of the identified molecules had a significantly higher fold change (mean $\log_2$(fold change) = 2.38) when compared to the genes that are solely induced by *S. aureus* supernatant (mean $\log_2$(fold change) = 1.50) (**Fig 8C**). For the down-regulated genes, the overlapping genes had a significantly lower fold change (mean $\log_2$(fold change) = −1.83) when compared to the genes solely repressed by *S. aureus* supernatant (mean $\log_2$(fold change) = −1.33) (**S8B Fig**).

We determined which molecules explained what proportion of the response to staphylococcal supernatant by generating an UpSet plot to show intersections of up-regulated genes in the supernatant with those among groups of the other conditions (**Figs 8D, S8C and S8D for expanded UpSet plots**). The largest intersections were between the shared up-regulated genes after addition of the combination of all molecules and the iron chelators DTPA/DFX (130 genes) and these conditions and StP (42 genes) (**Fig 8D**). These comparisons indicate that many of the genes induced by *S. aureus* supernatant are due to effects of exoproducts that induce iron-related pathways as well as zinc starvation; however, intersections were found between the up-regulated genes in *S. aureus* supernatant and each of the other conditions, showing their contribution to inducing distinct sensing pathways in *P. aeruginosa*. Taken together, we describe a model where *P. aeruginosa* senses the secreted staphylococcal molecules StP, iron chelators and unidentified proteins, and intermediate metabolites, and up-regulates metal starvation pathways, thereby producing PsP and pyoverdine, and metabolite uptake and catabolism pathways to use citrate and acetoin as carbon sources (**Fig 8E**).

## Discussion

In this study, we developed a systematic unbiased strategy using transcriptomics combined with genetics to determine the secreted exoproducts from one species that can be sensed by

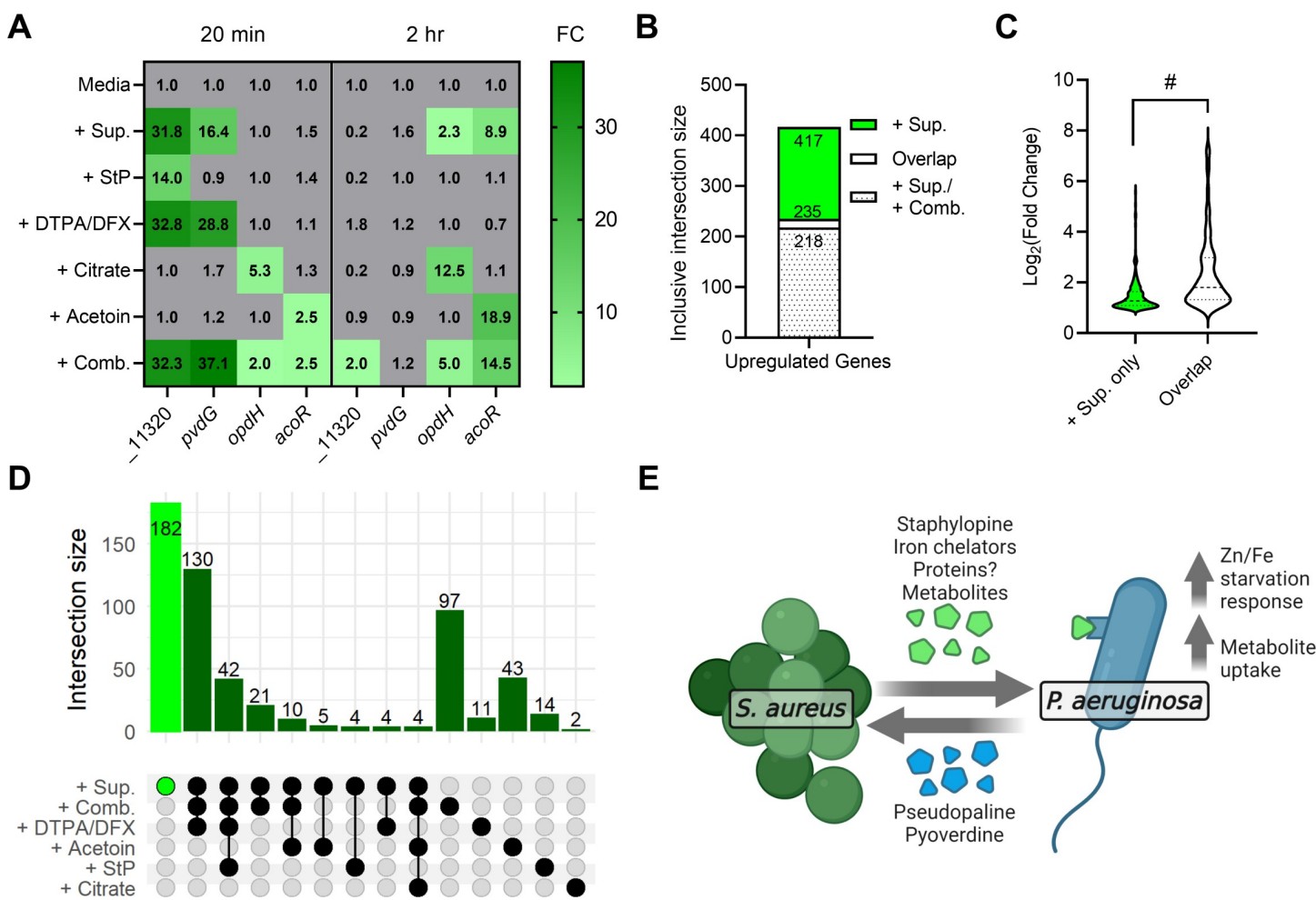

**Fig 8. Identified sensed products recapitulate a major part of the *P. aeruginosa* response to *S. aureus* supernatant. (A)** Heatmap of fold change of the transcripts PA14_11320, *pvdG*, *opdH*, and *acoR* after addition of *S. aureus* supernatant, the indicated molecules, or all molecules in combination (+ Comb.) over media control, after 20 minutes or 2 hours. **(B)** Inclusive intersection of all up-regulated genes after addition of supernatant (+ Sup.) versus at least one other condition (Overlap) or the combination of all molecules (+ Sup. / + Comb.). **(C)** Log$_2$ fold change of the up-regulated genes that are nonintersecting (+ Sup. only) or intersecting among supernatant and at least one other condition (Overlap). Medians (dashed lines) and first/third quartiles (dotted lines) are shown. Datasets were analyzed by a 2-tailed *t* test. #, *p* < 0.0001. **(D)** UpSet plot of exclusive intersections of up-regulated genes between addition of *S. aureus* supernatant and the indicated molecules. (See **S8 Fig** for full UpSet plot of all intersections.) **(E)** Model of *P. aeruginosa* sensing of *S. aureus* exoproducts. *S. aureus* secretes StP, iron chelators, intermediate metabolites, and possibly proteinaceous molecules that are sensed by *P. aeruginosa*. In response, *P. aeruginosa* up-regulates metal starvation and metabolite uptake pathways, which includes production of its own metal-binding molecules PsP and pyoverdine. The data underlying panels ABD and C can be found in Table M in S1 File and S1 Table, respectively. PsP, pseudopaline; StP, staphylopine.

another in a model two-species system. We used this approach to study the sensing of *S. aureus* by *P. aeruginosa*. Unlike other strategies that focus on a single molecule or response pathway of interest, our genome-scale approach revealed multiple staphylococcal factors that are sensed by *P. aeruginosa*, illustrating that interactions even between just two species can be complex and mediated by diverse molecules (**Fig 8E**). Our genome-wide screen also determined the mutations in *S. aureus* that affect the regulation, biosynthesis, and/or secretion of the sensed exoproducts. The sensed molecules we identified explain most of the *P. aeruginosa* transcriptional response to *S. aureus* in our system, suggesting that this systematic genome-wide approach can be applied to other two-species systems to comprehensively identify the underlying molecules that affect bacterial sensing in multispecies communities.

*P. aeruginosa* and *S. aureus* are commonly coisolated from lung infections in CF patients and together contribute to worse disease outcomes compared to monoinfections [25–27]. It is therefore imperative to better understand their interactions and the molecular mediators involved. Here, we found that *P. aeruginosa* senses multiple distinct *S. aureus* secreted molecules: the metallophore StP which affects zinc starvation and biofilm formation, unidentified molecules that affect pyoverdine production and/or deliver iron to *P. aeruginosa*, and the intermediate metabolites citrate and acetoin which are carbon sources for *P. aeruginosa*. Further, production and sensing of the sensed molecules was conserved in most clinical isolates of *S. aureus* and *P. aeruginosa*, respectively, suggesting that these pathways and interactions are common between these 2 species.

Staphylococcal exoproducts affected *P. aeruginosa* metal homeostasis, and zinc and iron chelators defined >45% of the transcriptional response to *S. aureus* supernatant (**Figs 8D and S8, and Table M in S1 File**). Zinc availability is required for virulence traits in both species, affecting *P. aeruginosa* motility, biofilm formation, and protease activity [61,62,72–74] and *S. aureus* biofilm formation and growth in vivo [53,75,76]. Interactions between *P. aeruginosa* and *S. aureus* in the context of zinc deprivation have been studied before in the presence of a neutrophil-derived metallophore, calprotectin, which binds zinc, manganese, nickel, and iron. Through metal chelation, calprotectin decreases antistaphylococcal antimicrobial production and inhibits extracellular protease-mediated lysis of *S. aureus*, thereby promoting coexistence between *P. aeruginosa* and *S. aureus* [60,61]. Moreover, it has been shown that the availability of zinc affects interactions between *P. aeruginosa* and the oral commensal species *Streptococcus sanguinis* SK36, likely due to zinc sequestration by *P. aeruginosa* [77]. We found that *S. aureus*–secreted StP led to the up-regulation of the biosynthesis genes of the metallophore PsP as well as other components of the Zur-dependent zinc starvation response in *P. aeruginosa* due to reduced availability of zinc. Additionally, StP inhibited *P. aeruginosa* biofilm formation and reduced *P. aeruginosa* antimicrobial killing of *S. aureus*. Our screen thus revealed a new interaction between *S. aureus* and *P. aeruginosa* through competition for, and modulation of, zinc availability by microbial opine metallophores. Zinc deprivation genes in *P. aeruginosa* are known to be induced in CF sputum [78–80], and our data suggest that competition with *S. aureus* may contribute to the induction.

*P. aeruginosa* and *S. aureus* are thought to compete for limited iron availability during infection via the secretion of iron-chelating siderophores and various iron uptake systems [28,81]. In a previous study, more *S. aureus* was recovered from coculture with a *P. aeruginosa* pyoverdine-deficient strain than WT, suggesting that pyoverdine is produced in the presence of *S. aureus* exoproducts and has an antagonistic effect on *S. aureus* [34]. However, other studies have shown a down-regulation of the *P. aeruginosa* iron starvation response in the presence of *S. aureus* or cell-free supernatant [35,70,82] and lysis of *S. aureus* liberates intracellular iron for acquisition by *P. aeruginosa* [35]. The differences between these studies could be due to variability in the iron content of the culture media as well as the *S. aureus* strains used. The results of our work support a complex effect of *S. aureus* on the *P. aeruginosa* iron response where on the whole *S. aureus* secreted products led to an induction of *P. aeruginosa* iron starvation responsive Fur- and PvdS-regulons, but *P. aeruginosa* obtains iron from specific staphylococcal chelators through xenosiderophore piracy, iron acquisition via uptake of a foreign siderophore [83]. Characterization of this intriguing iron piracy, and identification of the likely novel underlying molecules, will be the subject of a future study.

Through the transposon screen with P′*pvdG*, we also described the mutations in *S. aureus* that affect production of these molecules and hence iron-responsive pathways in *P. aeruginosa*, including in genes encoding metabolic enzymes, transcriptional regulators, and proteases. Particularly, supernatant from *S. aureus clpP* and *clpC* mutants led to decreased *pvdG* promoter

induction, but a large increase in relative chelation (**Fig 4B**). Clp proteases comprise the main proteolytic system in *S. aureus* and are known to affect iron homeostasis [84,85], and our data reinforce the idea that some *S. aureus* iron-related molecules may be direct or indirect targets of the Clp proteolysis machinery.

Our results also define a syntrophic relationship between *P. aeruginosa* and *S. aureus* where staphylococcal secreted intermediate metabolites citrate and acetoin can be used as carbon sources by *P. aeruginosa*. Costless secretion of valuable metabolites, especially of organic acids, is predicted to enable positive interspecies interactions through cross-feeding [86]. It was reported previously that *P. aeruginosa* can take up both metabolites as carbon sources [67,70,87,88], and acetoin was recently identified to participate in this cooperative interaction between *P. aeruginosa* and *S. aureus* in clinical coisolates [70], which validates our screening approach in determining sensed exoproducts. Previously, it was reported that in the presence of *P. aeruginosa* exoproducts, *S. aureus* secretes lactate, which is consumed by *P. aeruginosa* [34]. However, our *S. aureus* supernatant was obtained from monocultures and likely lacks production of this metabolite. Extracellular citrate has been detected in *S. aureus* cultures previously [89,90], and our work shows that this staphylococcal secreted metabolite can be sensed by *P. aeruginosa*, thus defining another cooperative interaction between these 2 coinfecting pathogens. Higher concentrations of citrate affect biofilm formation in *P. aeruginosa* [63] and extracellular citrate can stimulate quorum sensing in strains deficient for the LasR regulator [91]. Further, although citrate did not affect iron pathways in our study at the concentration found in *S. aureus* JE2 supernatant (**Table M in S1 File**), citrate can mediate iron uptake in *P. aeruginosa* [63,92], and it is possible that higher concentrations that facilitate iron uptake may exist in communities with these 2 species. Our genome-wide screen identified mutations in *S. aureus* that affect production and/or secretion of these sensed intermediate metabolites, which is not only important for better understanding these cooperative interspecies interactions and general metabolism but also has implications for the effects of these metabolites on *S. aureus* such as acetoin affecting stress resistance or citrate influencing virulence [90,93].

Previous studies have identified additional molecules that can be secreted by *S. aureus*, and are sensed by *P. aeruginosa*, such as *N*-acetyl glucosamine, which induces the *nag* operon, and autoinducer-2, which induces several virulence genes [36–40,42]. However, these products were not identified during *S. aureus* coculture and/or incubation with *S. aureus* supernatant, but rather with gram-positive bacterial communities. We did not see these response pathways being induced in our study upon exposure to *S. aureus* cell-free supernatant, possibly due to differences in growth medium, experimental conditions, and timing of gene expression measurements between the studies, and/or insufficient amounts of these products being present in 25% (v/v) *S. aureus* supernatant.

There are limitations and considerations to using our screening approach. First, our study was conducted in a defined medium to avoid introduction of molecules from another species into the 2-species system, such as culture media compositions that contain whole cell lysates, and it is important to consider that culture and growth conditions may affect either secreted molecule production and/or sensory responses. Next, the species being studied must have the necessary genetic tools available, such as reporter expression systems and random mutagenesis protocols, however these are becoming more widely accessible [94–96]. The identification of sensed products using our approach was more straightforward when there was a primary sensory molecule driving the response, such as in the case of StP, citrate, or acetoin, although fractionation of WT and mutant supernatants can reveal the molecules underlying multifactor-driven responses as well. Another consideration is that there is significant strain to strain variability in bacteria, and studies should therefore utilize a panel of strains to interrogate interspecies sensing. While our screen was carried out in a single strain of each species, we validated

that *S. aureus* clinical isolates from CF patients, including strains that were coisolated with *P. aeruginosa*, also produced molecules that induced the sensory pathways, albeit at varying levels, and that most *P. aeruginosa* clinical isolates also induced the response pathways.

Studies of interactions between *P. aeruginosa* and *S. aureus* have often focused on the effects of *P. aeruginosa* antimicrobials on staphylococcal growth and survival; however, it is an active question if these species, which are often coisolated, are more prone to competition or coexistence [31]. Here, by identifying the staphylococcal secreted products that are sensed by *P. aeruginosa*, we uncovered instances of both aspects: competition for metals, siderophore piracy, lifestyle alteration via inhibition of biofilm formation, and cooperation through release of intermediate metabolites used as carbon sources. Altering the production of these factors could potentially tip the balance in favor of a more beneficial or antagonistic interaction between these species.

## Materials and methods

### Bacterial strains and growth conditions

Bacterial strains utilized in this study are listed in S2 Table. *P. aeruginosa* UCBPP-PA14 [97] and *S. aureus* JE2 [98], other species listed, and their derivatives were grown in a modified M63 medium [99] (13.6 g·L$^{-1}$ KH$_2$PO$_4$, 2 g·L$^{-1}$ (NH$_4$)$_2$SO$_4$, 0.8 μM ferric citrate, 1 mM MgSO$_4$; pH adjusted to 7.0 with KOH) supplemented with 0.3% glucose, 1× ACGU solution (Teknova, Hollister, California, USA), 1× supplement EZ (Teknova), 0.1 ng·L$^{-1}$ biotin, and 2 ng·L$^{-1}$ nicotinamide, at 37˚C, shaken at 300 rpm, except where noted. *V. cholerae* was grown in M63 with the addition of 2% NaCl. For *P. aeruginosa* Δ*pvdJ* Δ*pchE* growth assays, 200 μM BIP (Sigma-Aldrich, St. Louis, Missouri, USA) was added to M63 medium. For sole carbon source growth assays, EZ and glucose were not added in the formulation. For each growth assay, *P. aeruginosa* overnight cultures were diluted to an OD$_{600}$ of 0.05 before monitoring growth. The NTML made up of 1,920 arrayed *bursa aurealis* transposon insertions in *S. aureus* JE2 was utilized for the transposon screen [50]. *S. aureus* and *P. aeruginosa* clinical isolates from the Cystic Fibrosis Foundation Isolate Core were selected from different patients, with 2 *S. aureus* isolates from the same patient, one of which was monoisolated and the other coisolated with *P. aeruginosa*. For cloning and strain construction, strains were routinely cultured in Luria Bertani (Miller) broth or on agar plates with 20 g·L$^{-1}$ agar. For selection, antibiotics were added as 50 μg·mL$^{-1}$ gentamicin, 25 μg·mL$^{-1}$ irgasan, 50 μg·mL$^{-1}$ carbenicillin, and/or 10 μg·mL$^{-1}$ erythromycin as needed. Other additives include StP (Santa Cruz Biotechnology, Dallas, Texas, USA), TPEN (Sigma-Aldrich), ZnSO$_4$·7H$_2$O (Sigma-Aldrich), NiCl$_2$·6H$_2$O (J. T. Baker, Phillipsburg, New Jersey, USA), CoCl$_2$·6H$_2$O (Sigma-Aldrich), DTPA (Sigma-Aldrich), deferoxamine mesylate salt (Sigma-Aldrich), and intermediate metabolites citric acid (Sigma-Aldrich), acetoin (Sigma-Aldrich), 2,3-butanediol (98% (v/v) solution, Sigma-Aldrich), and (S)-α-acetolactic acid potassium salt (Toronto Research Chemicals, North York, Ontario, Canada).

### Preparation of *S. aureus* cell-free supernatant

Overnight cultures of *S. aureus* strains were diluted to OD$_{600}$ of 0.05 in fresh media and grown for 24 hours in flasks (or tubes or 96-well plates for NTML library supernatant preparation). Bacterial cultures grown in flasks were centrifuged at 4,000 rpm for 20 minutes. The supernatant was applied to a Steriflip unit with a 0.2 μm polyethersulfone filter (MilliporeSigma, Burlington, Massachusetts, USA). For the resuspension of *S. aureus* exoproducts shown in S2 Fig, 5 mL of frozen flask supernatant was lyophilized overnight and then resuspended in the same volume of either media or water. For supernatants prepared from the NTML library in 1-mL

tube cultures or 96-well plates, culture was directly transferred to AcroPrep Advance 96-well 0.2 μm polyethersulfone filter plates (Pall, Port Washington, New York, USA) and centrifuged at 4,000 rpm for 10 minutes to collect the sterile-filtered flow through. Supernatants were stored at −30°C until use.

## RNA extraction and library preparation

*P. aeruginosa* overnight cultures were diluted to $OD_{600}$ of 0.05 in fresh media in flasks and grown to $OD_{600}$ of 0.50. Two milliliters were collected for RNA extraction immediately before addition of 5 mL (25% v/v) of either fresh media, supernatant, or additives in media and collected after 20 minutes, 1 hour, and 2 hours for the data described in Fig 1 and after 20 minutes and 2 hours for the data described in Fig 8. Removed culture was mixed with 4 mL of RNAprotect Bacteria Reagent (QIAGEN, Hilden, Germany) for stabilization and incubated for 5 minutes at room temperature before centrifugation at 4,000 rpm for 10 minutes. Supernatants were completely removed before storage of the pellets at −80°C. RNA was extracted using the Total RNA Purification Plus Kit (Norgen, Thorold, Ontario, Canada) according to the manufacturer's instructions for gram-negative bacteria. The extracted RNA was subjected to an additional genomic DNA removal by DNase I treatment in solution using the TURBO DNA-free Kit (Invitrogen, Waltham, Massachusetts, USA) and checked by PCR for the absence of contaminating DNA. Integrity of the RNA preparation was confirmed by running on an agarose gel and observing intact rRNA bands. Next, rRNA was removed using the Ribo-Zero Depletion Kit for Gram Negative Bacteria (Illumina, San Diego, California, USA) or ribo-POOLs (siTOOLs Biotech, Planegg, Germany) before library preparation with the NEBNext Ultra II Directional RNA Library Prep Kit for Illumina (New England Biolabs, Ipswich, Massachusetts, USA). The sequencing was performed at the Center for Cancer Research Genomics Core Facility. Two biological replicates were performed.

## RNA-seq analysis

The sequencing files were processed with Cutadapt [100] and Trimmomatic [101]. Alignment to the *P. aeruginosa* UCBPP-PA14 genome (NCBI) and pairwise comparisons were made using Rockhopper (Wellesley College) [102, 103]. Up-regulated and down-regulated genes were based on transcripts that had $p < 0.05$ and $\log_2$ fold change $\geq 1$ or $\leq -1$. Venn diagrams were generated using matplotlib_venn package with venn3 using Python [104]. All up-regulated genes were evaluated for increasing fold change over time. UpSet plots were generated with ComplexUpset using R [105]. For UpSet plots, all up-regulated genes at 20 minutes and 2 hours were combined and compared between datasets.

## Construction of *P. aeruginosa* promoter-reporter strains and mutants

Bacterial strains and plasmids are listed in S2 Table. Promoter sequences were amplified using the primers listed in S3 Table. The amplified fragments were cloned by Gibson Assembly (New England Biolabs) into SpeI- and XhoI-digested pSEK109 (pLB3208), provided by the Dietrich lab [106]. Plasmids were transformed into *E. coli* S17-1 λ-pir and confirmed by sequencing. Next, bi-parental conjugations were conducted as described previously between *E. coli* donor cells and *P. aeruginosa* recipients [20]. For each conjugation, cells were collected and plated on LB + irgasan + gentamicin plates. For PA14, the integration was then unmarked using Flp-FRT recombination by mating with an *E. coli* donor strain harboring the pFLP2 plasmid, selecting on LB + irgasan + carbenicillin, and further curing of this plasmid by selecting on LB plates (without NaCl) + 10% sucrose [107]. All promoter-reporter strains were confirmed by PCR with the forward primer for the promoter and the reverse primer internal to

mScarlet Pa094. For mutant construction, homologous downstream and upstream arms of *cnt* genes were amplified using the primers listed in S3 Table. The amplified fragments were cloned into pDONRPEX18Gm *attP* sites using the Gateway BP Clonase II Enzyme mix (Thermo Fisher, Waltham, Massachusetts, USA). Plasmids were transformed into *E. coli* S17-1 λ-pir and confirmed by sequencing prior to conjugation as described above. Conjugants were streaked onto LB plates (without NaCl) + 10% sucrose and then tested for gentamicin resistance. Gentamicin-sensitive strains were tested for the deletion by PCR and sequencing.

## Plate reporter assay

*P. aeruginosa* overnight cultures were diluted to $OD_{600}$ of 0.05 in fresh media in flasks and grown to $OD_{600}$ of 0.5. At this time, 150 μL of culture was added to wells of a 96-well clear, flat-bottom polystyrene plate with 50 μL fresh media (control) or *S. aureus* supernatant with or without additives as described. Plates were incubated at 37˚C with continuous orbital shaking at 807 cycles per minute (cpm) in a Synergy H1 microplate reader (BioTek, Winooski, Vermont, USA) with fluorescence and optical density measurements every 30 minutes (mScarlet: 565 nm excitation and 600 nm emission; pyoverdine: 405 nm excitation and 460 nm emission [108]; optical density at 600 nm). Measurements for duplicate wells were averaged, and the fluorescence was normalized to the growth (relative fluorescence units divided by the $OD_{600}$). Slopes were calculated from 1.5 to 5 hours (promoters of PA14_11320, *cntO*, PA14_73020, *pvdG*, *opdH*, and *acoR*) or 1 to 4 hours (promoter of *pvdG* and pyoverdine). To calculate the z-score, for each individual experiment (96-well plate), the mean slope and standard deviation were determined. Each sample replicate z-score is the number of standard deviations away from the mean, calculated using the formula $z = (x-\mu)/\sigma$, where x is the sample slope, μ is the mean slope, and σ the standard deviation.

## GO enrichment analysis

For *P. aeruginosa* PA14 pathway analysis, the open reading frame designations for the corresponding PAO1 orthologs of *P. aeruginosa* UCBPP-PA14 genes were obtained using the *Pseudomonas* Genome Database (www.pseudomonas.com/rbbh/pairs) [109]. Similarly, for *S. aureus* JE2 pathway analysis, the open reading frame designations for the corresponding NCTC8325 orthologs of *S. aureus* JE2 USA300_FPR3757 genes were obtained using the AureoWiki repository (http://aureowiki.med.uni-griefswalk.de/Downloads) [110]. The list of designations from the RNA-seq analysis (*P. aeruginosa*) or selected transposon mutants (*S. aureus*) were analyzed at the GO Resource (www.geneontology.org) by the PANTHER Overrepresentation Test (released 20210224) (Annotation Version and Release Date: GO Ontology database DOI: 10.5281/zenodo.5228828 Released 2021-08-18) or (released 20200728) (Annotation Version and Release Date: GO Ontology database DOI: 10.5281/zenodo.4081749 Released 2020-10-09), respectively, for enriched biological processes by Fisher exact test with Bonferroni correction.

## Biochemical fractionation

Five milliliters of *S. aureus* supernatant were Chelex-treated according to the manufacturer's instructions for 1 hour at room temperature with agitation, then frozen and lyophilized before resuspension in 500 μL of filtered water. The resuspension was separated using a Superdex 30 10/300 GL size exclusion column (GE Healthcare) with filtered water as buffer at a flow rate of 0.7 mL·min$^{-1}$ for 1.5 column volumes at 2.6 MPa for a total of 36.5 mL with 18 fractions collected after every 2 mL. Fractions were frozen, lyophilized, and resuspended in 500 μL of water. For protease digestion, 60 μL of each fraction was treated with 1 μL of thermolabile proteinase K (New England Biolabs) for 1 hour at 37˚C followed by inactivation for 10 minutes at 55˚C.

## Biofilm formation assay

*P. aeruginosa* was grown overnight in modified M63 medium. An overnight culture was diluted to a starting $OD_{600}$ of 0.05 in fresh medium mixed with the indicated additives. Moreover, 100 μL of culture was added to 8 replicate wells of a 96-well plate. Biofilms were allowed to form in static conditions for 24 hours at 37°C and assessed for biofilm formation using crystal violet staining [111].

## Supernatant survival assay

To prepare cell-free supernatant, *P. aeruginosa* was inoculated from an overnight culture at an $OD_{600}$ of 0.05 and allowed to grow for 16 hours in medium with the indicated additives. The culture was centrifuged, and the supernatant was filtered with a Steriflip unit with a 0.2 μm polyethersulfone filter (MilliporeSigma). For the survival assay, *S. aureus* was inoculated from an overnight culture at an $OD_{600}$ of 0.05 and grown to an $OD_{600}$ of 0.5. Next, 500 μL of culture was added to 500 μL (50% v/v final concentration) of either *P. aeruginosa* cell-free supernatant or M63 salts control. After 16 hours, *S. aureus* was plated on LB agar plates and enumerated after growth.

## Chromeazurol S (CAS) assay for iron chelation

Solutions were prepared exactly as described and assay was performed as described with some changes [112]. Moreover, 100 microliters of M63 media reference or 75 μL M63 with 25 μL of *S. aureus* supernatant (1X concentration) were added to a 96-well plate and mixed with 100 μL CAS shuttle solution. Plates were incubated for 1 hour at room temperature. The absorbance was measured at $OD_{630}$. The absorbance ratio was calculated by dividing the average absorbance for the sample by the absorbance of the media reference. Next, this ratio was normalized to a standard log-based curve of WT supernatant to obtain the relative chelation.

## Citrate measurements

Citrate concentrations were measured in *S. aureus* supernatant using the Citric Acid Assay Kit (Megazyme, Bray, Ireland) according to the microplate assay instructions. Absorbances after the enzymatic reaction were compared to a standard log-based curve of citric acid concentrations to determine the concentration in supernatant.

## Acetoin and -acetolactate measurements

Acetoin was measured by the Voges–Proskauer test adapted for microtiter plates essentially as described [70,113]. In order, 35 μL 0.5% (m/v) creatine (Sigma-Aldrich, St. Louis, Missouri, USA), 50 μL 5% (m/v) -naphthol (Sigma), 50 μL 40% (v/v) KOH, and 50 μL *S. aureus* supernatant were added to each well. The reaction was incubated at room temperature for 15 minutes and absorbance was measured at $OD_{560}$. Concentrations were calculated based on a linear standard curve. To quantify -acetolactate, 5 μl 2M $H_2SO_4$ was added to supernatant and incubated for 15 minutes at 60°C for decarboxylation of α-acetolactate to acetoin before adding to the above reagents as described [114]. Due to precipitation in the wells, each mixture was transferred to a new well before reading absorbance.

## Quantification of 2,3 butanediol by mass spectrometry

Stock solution of 50 mM 2,3-butanediol (Sigma-Aldrich) and calibration standards from 0 to 40 μM were prepared in water. The isotopic standard (IS) solution of 5 μM 2,3-butanediol (Toronto Research Chemicals) was prepared in water. Moreover, 2,3-butanediol

concentrations in the samples were determined by LC-MS/MS after derivatization with tri-chloroacetyl isocyanate (Sigma-Aldrich) according to Chen and colleagues [115] with slight modification. In a 1.5 mL plastic vial, 5 μL of standard or sample was mixed with 5 μL IS, 250 μL acetonitrile, and 20 μL trichloroacetyl isocyanate. After 5 minutes, 250 μL water was added to the vials and the mixture were analyzed by liquid chromatography-mass spectrometry LC-MS/MS. LC were performed with a Shimadzu 20AC-XR system with a $2.1 \times 100$ mm, Cortecs C8 column (Waters, Milford, Massachusetts, USA). Mobile phase A was 10 mM ammonium formate with 0.15% (v/v) formic acid in water and mobile phase B was methanol. The flow rate was 300 μL·min$^{-1}$, and the injection volume was 5 μL. Furthermore, 2,3-butanediol was eluted from the column using a gradient (0 to 0.2 min/55% (v/v) B; 5 min/70% (v/v) B; 5.1 to 6 min/95% (v/v) B; 6.1 to 8 min/55% (v/v) B). MS/MS was performed with a TSQ Vantage triple quadrupole mass spectrometer (Thermo Fisher Scientific, Waltham, Massachusetts, USA) operating in selected reaction monitoring mode with positive ionization. The derivatized peaks were detected using the following mass-to-charge ratio (*m/z*) precursor > product ions: 2,3-butanediol (484 > 262); 2,3-butanediol-d8 (492 > 270). Moreover, 2,3-butanediol concentrations in the samples were determined by a linear calibration curve with 1/x weighting generated by the Thermo Xcalibur software. The curve was constructed by plotting the peak area ratios versus standard concentrations. The peak area ratio was calculated by dividing the peak area of 2,3-butanediol by the peak area of the IS.

## Accession number(s)

The RNA-seq data have been deposited at NCBI Gene Expression Omnibus (GEO) (https://www.ncbi.nlm.nih.gov/geo) under accession numbers GSE185963 and GSE186138.

## Supporting information

**S1 Fig. Heatmap of up-regulated *P. aeruginosa* genes that increased in fold change over time upon addition of *S. aureus* supernatant.** The log$_2$ fold changes for all up-regulated genes with increasing fold change over time from 20 minutes to 1 hour to 2 hours are shown. Gray, log$_2$ fold change < 1. The data underlying this figure can be found in Tables A and B in S1 File.
(TIF)

**S2 Fig. Characterization of the effect of *S. aureus* cell-free supernatant pH and nutrient levels on growth and promoter responses in *P. aeruginosa*.** **(A)** The pH of 100% *S. aureus* supernatant and 25% (v/v) *S. aureus* supernatant in medium was measured. **(B)** Growth of *P. aeruginosa* was measured after exposure to 25% *S. aureus* supernatant or media control during the plate reader assay. **(C)** RFUs of mScarlet normalized to OD$_{600}$ over time after exposure to media control, the salts from the media, *S. aureus* supernatant, or lyophilized *S. aureus* supernatant resuspended in either media or water, in the indicated promoter-reporter strains. Slope calculated from 1.5 to 5 hours (promoters of PA14_11320, *opdH*, and *acoR*) or 1 to 4 hours (promoter of *pvdG*). Datasets were analyzed by 1-way ANOVA with Dunnett test for multiple comparisons to the media control. Data shown for all panels are the means of 3 independent biological replicates, including 3 separate supernatant collections in A. Shaded regions and error bars denote the SD. *, $p < 0.05$; **, $p < 0.01$; ***, $p < 0.001$; #, $p < 0.0001$. The data underlying all panels can be found in S1 Table. RFU, relative fluorescence unit; SD, standard deviation.
(TIF)

**S3 Fig. StP and zinc levels affect *S. aureus* and *P. aeruginosa* physiology.** **(A)** WT *S. aureus* and NTML transposon mutants were grown in media with or without the addition of the zinc

chelator TPEN at the concentrations indicated. Growth was measured at $OD_{600}$ after 24 hours. Datasets were analyzed by a 2-way ANOVA with Dunnett test for multiple comparisons to the respective WT value. **(B, C)** RFU of mScarlet normalized to $OD_{600}$ over time after exposure to media control or *S. aureus* supernatant and/or addition of the indicated concentrations of **(B)** TPEN or **(C)** zinc, nickel, and cobalt in the indicated *P. aeruginosa* promoter-reporter strains (promoters of PA14_11320, *cntO*, or *dksA2*). Datasets were analyzed by a 2-way ANOVA with Tukey test for multiple comparisons. Statistics shown represent tests comparing **(B)** the respective conditions/promoters with and without TPEN and **(C)** all treatments to the respective supernatant addition. **(D)** *S. aureus* cultures were grown in culture with 50% medium salts (Control) or cell-free supernatants of *P. aeruginosa* grown in the presence of the indicated additives. Colony-forming units per milliliter of *S. aureus* were calculated after 16 hours of growth. Datasets were analyzed by 1-way ANOVA with Dunnett test for multiple comparisons to the *S. aureus* grown in supernatant from *P. aeruginosa* grown in media without additions. Data shown for all panels are the means of 3 independent biological replicates, including 3 separate supernatant collections in D. The error bars denote the SD. *, $p < 0.05$; **, $p < 0.01$; ***, $p < 0.001$; #, $p < 0.0001$. The data underlying all panels can be found in S1 Table. NTML, Nebraska Transposon Mutant Library; RFU, relative fluorescence unit; SD, standard deviation; StP, staphylopine; WT, wild type.
(TIF)

**S4 Fig. Iron chelators affect induction of the *pvdG* promoter and iron delivery to *P. aeruginosa*. (A)** Chelation by different amounts of *S. aureus* supernatant was measured by CAS assay to construct a standard curve to calculate relative chelation. **(B)** RFU of mScarlet expressed from the promoter of *pvdG* normalized to $OD_{600}$ over time after exposure to media control, *S. aureus* supernatant, or the indicated concentrations of the iron chelators DFX and DTPA. Data shown from 3 independent replicates. **(C)** Slope of $OD_{600}$ over time from 1 to 8 hours of *P. aeruginosa* Δ*pvdJ* Δ*pchE* strain in medium control or medium with 200 μM BIP with the addition of whole WT *S. aureus* supernatant, media, or the iron chelators DFX or DTPA. Error bars denote the SD. The data underlying all panels can be found in S1 Table. CAS, chromeazurol S; DFX, deferoxamine; DTPA, diethylene triamine penta-acetic acid; RFU, relative fluorescence unit; SD, standard deviation; WT, wild type.
(TIF)

**S5 Fig. The *acoR* promoter is induced by intermediate metabolites acetoin, 2,3-butanediol, and α-acetolactate but not citrate. (A)** RFU of mScarlet expressed from the *acoR* promoter normalized to $OD_{600}$ over time after exposure to media control, *S. aureus* supernatant, or addition of the indicated concentrations of each metabolite. **(B)** Quantification of 2,3-butanediol and α-acetolactate in *S. aureus* supernatant. Data shown for all panels is from 3 independent replicates. Error bars denote the SD. The data underlying all panels can be found in S1 Table. RFU, relative fluorescence unit; SD, standard deviation.
(TIF)

**S6 Fig. *P. aeruginosa* clinical isolates induce responses to *S. aureus* supernatant.** RFU of mScarlet normalized to $OD_{600}$ over time after exposure to *S. aureus* supernatant or media control from the indicated promoter-reporter constructs in *P. aeruginosa* PA14 and clinical isolate strains CF17, CF33, CF72, and CF104. Slope calculated from 1.5 to 5 hours. Data shown for all panels are the means of 3 independent biological replicates. Datasets were analyzed by unpaired *t* tests to the respective media control. The error bars denote the SD. *, $p < 0.05$; **, $p < 0.01$; ***, $p < 0.001$; #, $p < 0.0001$. The data underlying all panels can be found in

S1 Table. RFU, relative fluorescence unit; SD, standard deviation.
(TIF)

**S7 Fig. Secreted products from diverse species induce response pathways in *P. aeruginosa*.**
RFU of mScarlet expressed from the indicated promoter or Pvd normalized to $OD_{600}$ over time (promoter of PA14_11320, *opdH*, and *acoR* calculated from 1.5 to 5 hours; Pvd calculated from 1 to 4 hours) after exposure to media control, *S. aureus* supernatant, or supernatant from the indicated species: *Bacillus subtilis*, *Staphylococcus epidermidis*, *Burkholderia cenocepacia*, *Escherichia coli*, *Klebsiella pneumoniae*, *Salmonella enterica* Typhimurium, *Stenotrophomonas maltophilia*, and *Vibrio cholerae*. Data shown from at least 3 independent replicates. Error bars denote the SD. Datasets were analyzed by 1-way ANOVA with Dunnett test for multiple comparisons to the control. *, $p < 0.05$; **, $p < 0.01$; ***, $p < 0.001$; #, $p < 0.0001$. The data underlying all panels can be found in S1 Table. Pvd, pyoverdine; RFU, relative fluorescence unit; SD, standard deviation.
(TIF)

**S8 Fig. Identified sensed products explain different proportions of the *P. aeruginosa* response to *S. aureus* supernatant. (A)** Inclusive intersection of all down-regulated genes after addition of supernatant (+ Sup.) versus at least one other condition (Overlap) or the combination of all molecules (+ Sup. / + Comb.). **(B)** $Log_2$ fold change of the down-regulated genes that are nonintersecting (+ Sup. only) or intersecting among supernatant and at least one other condition (Overlap). Medians (dashed lines) and first/third quartiles (dotted lines) are shown. Datasets were analyzed by a 2-tailed *t* test. #, $p < 0.0001$. **(C, D)** UpSet plot showing all exclusive intersections of (C) up-regulated or (D) down-regulated genes at 20 minutes and 2 hours after addition of *S. aureus* supernatant, the indicated products, or the combination of all products (+ Comb.). All intersections are shown. The data underlying panels ACD and B can be found in Table M in S1 File and S1 Table, respectively.
(TIF)

**S1 File. Excel file containing Tables A–M.**
(XLSX)

**S1 Table. Excel file containing numerical data underlying Figs 2B, 3B, 3C,3D, 3E, 3F, 3G, 4C, 4D, 4E, 5D, 6B, 6E, 7A, 7B, 7C, 7D, 8C, S2A, S2B, S2C, S3A, S3B, S3C, S3D, S4A, S4B, S4C, S5A, S5B, S6, S7 and S8B.**
(XLSX)

**S2 Table. Bacterial strains and plasmids used in this study.**
(PDF)

**S3 Table. Primers used in this study.**
(PDF)

## Acknowledgments

We would like to acknowledge the Center for Cancer Research Genomics Core for RNA-sequencing, King Chan at the Protein Characterization Laboratory at Frederick National Laboratory for Cancer Research for quantifying 2,3-butanediol, the Cystic Fibrosis Foundation Isolate Core for providing clinical isolate strains, the Dietrich lab (Columbia University) for providing the pSEK109 (pLD3208) plasmid, and the Adhya and Ramamurthi labs (NIH) for providing isolates of different bacterial species. We thank Susan Gottesman, Kumaran Ramamurthi, Gisela Storz, and Saaed Tavazoie for comments on the manuscript, and

members of the Gottesman, Ramamurthi, and Khare labs for discussion and feedback throughout this study. This work utilized the computational resources of the NIH High Performance Computing Biowulf cluster (http://hpc.nih.gov). Diagrams in Figs 1, 3, 5, 6, and 8 were created with BioRender (http://biorender.com).

## Author Contributions

**Conceptualization:** Tiffany M. Zarrella, Anupama Khare.

**Data curation:** Tiffany M. Zarrella, Anupama Khare.

**Formal analysis:** Tiffany M. Zarrella, Anupama Khare.

**Funding acquisition:** Tiffany M. Zarrella, Anupama Khare.

**Investigation:** Tiffany M. Zarrella.

**Methodology:** Tiffany M. Zarrella, Anupama Khare.

**Project administration:** Anupama Khare.

**Resources:** Anupama Khare.

**Supervision:** Anupama Khare.

**Visualization:** Tiffany M. Zarrella, Anupama Khare.

**Writing – original draft:** Tiffany M. Zarrella, Anupama Khare.

**Writing – review & editing:** Tiffany M. Zarrella, Anupama Khare.

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
