## [Editor Report · Decision Letter 0]

6 Dec 2021

Dear Dr Khare, 

Thank you for submitting your manuscript entitled "Systematic identification of molecular mediators of interspecies sensing in a two-species bacterial community" for consideration as a Research Article by PLOS Biology.

Your manuscript has now been evaluated by the PLOS Biology editorial staff, as well as by an academic editor with relevant expertise, and I'm writing to let you know that we would like to send your submission out for external peer review.

Once your full submission is complete, your paper will undergo a series of checks in preparation for peer review. Once your manuscript has passed the checks it will be sent out for review. To provide the metadata for your submission, please Login to Editorial Manager (https://www.editorialmanager.com/pbiology) within two working days, i.e. by Dec 08 2021 11:59PM.

If your manuscript has been previously reviewed at another journal, PLOS Biology is willing to work with those reviews in order to avoid re-starting the process. Submission of the previous reviews is entirely optional and our ability to use them effectively will depend on the willingness of the previous journal to confirm the content of the reports and share the reviewer identities. Please note that we reserve the right to invite additional reviewers if we consider that additional/independent reviewers are needed, although we aim to avoid this as far as possible. In our experience, working with previous reviews does save time. 

If you would like to send previous reviewer reports to us, please email me at rroberts@plos.org to let me know, including the name of the previous journal and the manuscript ID the study was given, as well as attaching a point-by-point response to reviewers that details how you have or plan to address the reviewers' concerns. 

Given the disruptions resulting from the ongoing COVID-19 pandemic, please expect some delays in the editorial process. We apologise in advance for any inconvenience caused and will do our best to minimize impact as far as possible.

Kind regards,

Roli Roberts

Roland Roberts

Senior Editor

PLOS Biology

rroberts@plos.org

---

## [Decision Letter · Decision Letter 1]

25 Jan 2022

Dear Dr Khare,

Thank you for submitting your manuscript "Systematic identification of molecular mediators of interspecies sensing in a two-species bacterial community" for consideration as a Research Article at PLOS Biology. Your manuscript has been evaluated by the PLOS Biology editors, an Academic Editor with relevant expertise, and by three independent reviewers. Please accept my apologies for the delay over the holiday period.

You'll see that while all three reviewers are impressed by the scale of your study, and reviewers #2 and #3 are overall broadly positive, reviewer #1 raises significant concerns about the possibility that you are merely observing a generic starvation response, and the limited connection to clinical situations. I discussed this discordance of opinion with the Academic Editor, who said that you should "address the critical comments of Reviewer #1 by performing a few additional experiments, particularly to rule out that what they see is a general starvation response. This should be relatively straightforward... In principle, even testing few clinical strains to confirm that they exhibit similar responses as PA14 (without redoing the screen) should not be that difficult." I hope these comments are useful in helping you to decide how to revise your manuscript.

In light of the reviews (below), we will not be able to accept the current version of the manuscript, but we would welcome re-submission of a much-revised version that takes into account the reviewers' comments. We cannot make any decision about publication until we have seen the revised manuscript and your response to the reviewers' comments. Your revised manuscript is also likely to be sent for further evaluation by the reviewers.

We expect to receive your revised manuscript within 3 months. 

**IMPORTANT - SUBMITTING YOUR REVISION**

*Re-submission Checklist*

*Published Peer Review*

*PLOS Data Policy*

*Blot and Gel Data Policy*

Sincerely,

Roli Roberts

Roland Roberts

Senior Editor

PLOS Biology

rroberts@plos.org

REVIEWERS' COMMENTS:

Reviewer #1:

In the manuscript Systematic identification of molecular mediators of interspecies sensing in a two-species bacterial community, Zarrela and Khare studied the metabolites secreted by S. aureus that influenced the metabolism of P. aeruginosa using a systems biology approach. This referee acknowledged the amount of work that is presented in this work. However, I have important reservations related to the study design and the experimental outcome of this work:

-The strain selection. Patients with cystic fibrosis usually develop long-term infections. S. aureus and P. aeruginosa are often found in the airways of these patients. Studies show higher prevalence of P. aeruginosa than S. aureus in older CF patients as it is known that P. aeruginosa replaces S. aureus during the course of the infection. More recently, it has been shown that that the two species are able to coexist in the lungs of CF patients. Because of this, the interspecies competition/coexistence of these two bacterial species is one of the most studied case of multispecies community in microbiology. Therefore, it is important to study these interactions in an environmental niche the closely mimics the CF disease. The strain of P. aeruginosa used in this study is PA14, a classical isolate for laboratory studies which is not a CF strain. P. aeruginosa rapidly adapts to the host immune responses and antibiotic treatments of CF patients. This adaptation involves the accumulation of genetic mutations that alter genes expression and phenotypes of P. aeruginosa (Smith et al., 2006; Marvig et al., 2015a; Winstanley et al., 2016; La Rosa et al., 2019). Therefore, CF strains of P. aeruginosa differ from PA14, as they are specifically adapted to persist in a CF infection niche. In the case of the S. aureus strain, the authors did not use a clinical isolate but a genetically-modified S. aureus strain JE2. This strain JE2 or any other strain with a similar genetic background will not be found in any infection, as it results from genetic manipulation in a laboratory. The strain JE2 does not have the plasmids which confer to MRSA strains their multi-drug resistance profile as well as gene expression modifications. Overall, the use of these PA and SA strains does not represent the interaction niche of a CF microbial community.

-The methodology. The authors used cell-free exhausted supernatant from S. aureus cultures to supplement P. aeruginosa cultures. They authors claimed that, in 25% V/V supplemented P. aeruginosa cultures, this bacterium will respond to the presence of S. aureus secreted biomolecules. This is partially true. The most important effect associated with this medium supplementation will be a starvation/stress response that is consequence of 25% nutrient deprivation. Adding 25% of exhausted growth medium to the culture will affect growth of P. aeruginosa, irrespective on whether the exhausted medium was obtained from a S. aureus culture or any other species. Using this methodology, the three most important responses of P. aeruginosa will be the following: 1) a substantial starvation response that is consequence of a 25% nutrient deprivation., This will induce the genes related to the TCA cycle, to enable the bacterium to use alternative metabolic sources, such as citrate (see the following point). The starvation response also includes low availability of metal ions, which induces the secretion of metal chelators. 2) A stress response that is consequence of a pH variation, which includes the expression of genes involved in the ROS response. 

To avoid these effects, the cultures supplementation studies are usually carried out using exhausted medium corrected in pH and nutrient availability, in such they are comparable to that of other regular growth media. 

- The bacterial response. The authors claimed that the strongest P. aeruginosa response to S. aureus cell-free supernatant was represented by four pathways: Zn-deprivation, Fe-deprivation, TCA uptake, and acetoin catabolism. This is however, a classical starvation response which likely is a consequence of the 25% nutrient deprivation of supplemented cultures. in other words, the authors will obtain a comparable response by complementing PA cultures with spent media from any other cultures or even with low-pH water supplemented with alternative metabolic sources such as citrate or acetate. This is because, during exponential growth in laboratory cultures, carbohydrates are quite available and P. aeruginosa uses respiration to grow (via TCA cycle) until the concentration of carbohydrates decreases. Only then do the bacteria rely on other metabolic sources, such as citrate or acetate, to feed the TCA to obtain energy. This involves an induction of the TCA cycle to maintain an effective growth, as citrate or acetate are not energetically as efficient as are the carbohydrates. This occurs indeed during starvation conditions. The same applies to the deprivation of metal ions, which are essential for many enzymatic reactions. A 25% nutrient deprivation in the supplemented cultures causes a defect metal ions availability; bacteria reacted to this effect by inducing the production of metal chelators.

Overall, I think the system biology approach presented in this work is quite interesting but the experimental outcome does not allow to obtain important information to understand the interaction/coexistence of P. aeruginosa and S. aureus in the lung of CF patients.

Reviewer #2:

In this paper by Zarella and Khare, the authors dissect multiple stimuli that influence P. aeruginosa that are present in S. aureus supernatants. The paper summarizes a lot of work and is pitched as a system for the discovery of metabolites mediating interaction. The major factors in S. aureus supernatants that affect the P. aeruginosa transcriptome include the zincophore staphylopine, iron binding factors, citrate and acetoin. The work melds the use of transcriptomics and fractionation to genetic screens of a mutant S. aureus library to identify mutants that no longer trigger these responses. I think that the approach of combined transcriptomics with genetic screens has been well-used, a fact that may be underrecognized in the introduction/set-up of the paper. 

The major contribution of this paper is in the final figures in which the authors look at the additive effects of the different S. aureus stimuli on the overall P. aeruginosa response which is an exciting next step in microbe-microbe interactions research and show that these responses are observed even if other clinical strains are used. I am really impressed with Figures 7 (clinical isolates) and in particular, Fig. 8. Line 105-8-the approach aims to quantify the fraction of the transcriptional response that can be explained by the different small molecule stimuli. In the end, about ~50% of the response (97 genes of 184 genes) to supernatant could be explained by four different stimuli. This nicely highlights what was achieved and what remains to be explained. 

Specific comments: 

1. Table S5. It would be helpful if the cnt mutants that were identified in the screen in Table S5 were identified. (I was not able to determine which genes were identified in the screen using publicly available databases.) It is not clear that the cnt mutants were the top candidates in the screen or if they were pulled from the collection based on their known function and analyzed in a targeted fashion in Fig. 2. The top hits in the table did not seem to be involved in StP biosynthesis. 

2. Please include the specifics for the z-score transformations. 

3. The data in Fig. 5ABC and Fig. 6ACD seems peripheral and in the end, particularly for Fig. 6, the data didn't strongly support a correlation between levels. I suggest moving to the supplement (despite the fact that they represent a lot of work). The inclusion of these data make this data-rich paper harder to follow. 

4. Please include more specifics on the z-score calculations. 

5. Some references for these molecules in other P. aeruginosa interactions: 

a) Please include mention of Mould et al. as evidence for extracellular citrate response inducing a P.a. response and the enhancement of this interaction with iron siderophores. 

b) More clear reference to work by Skaar et al. on the topic of zincophores in P.a.-Staph interactions. 

c) Recognize published interactions with S.a. such as P.a. production of HQNO and siderophores in co-culture not mentioned (or not mentioned clearly) by O'Toole and colleagues. Citrate and HAQs are particularly important given that they identify metal scavenging as one of the signals induced by Staph sup.

6. Confirm that all figure legends specify which promoter fusion was used-in at least two places it was necessary to look in the results for this info. 

7. Lines 187-188 "indicating that similar concentrations are likely present in staphylococcal supernatant." This is a strong statement. It would be more appropriate to say that this concentration is sufficient to induce to levels noted with supernatant induction. (This is especially in light of data showing that stp mutant supernatant still induced higher levels of the zinc responsive promoter than media control to suggest the presence of an additional factor.)

8. Line 194. The data on P. aeruginosa cnt mutants is not relevant to the response to Staph supernatants. These data should be moved to the supplement or removed. 

9. Throughout, for the phenotype assays, it would be useful to know where there are differences in growth rather than only showing the OD normalized data. The data in Fig. 2 show a delay rather than a lack of induction for zinc and iron responses which makes for a very strong response in the 1-4 hour window. Based on the pattern for data in Fig. 2A for 11320, it seems that the overnight cultures were limited for zinc. Please note when differences in growth are observed. 

Reviewer #3: 

Summary

The authors have developed an elegant approach to identify both the interspecies signals and the mechanisms of sensing in a two-member bacterial community by combining transcriptomics, genetic screens, and proteomics approaches. The authors utilized the well-characterized pair P. aeruginosa and S. aureus, which allowed for the identification of previously published pathways, helping to validate the utility of the approach, but also previously unidentified interactions. The authors also followed up briefly on the four major pathways identified (zinc and iron deprivation, TCA intermediate uptake, and acetoin metabolism) with genetic and phenotypic studies yielding deeper insight into interactions between these important pathogens. The paper is extremely well-written and the figures are beautifully designed. I have only a few very minor suggestions below. 

1. The authors remark that this approach can exhaustively reveal the molecules that lead to response of a foreign species. While I am impressed with the authors' multi-technique approach, I wonder one validates that it is exhaustive?

2. The clarity of Figure 3 could be improved with more detailed labeling on the Figure and in the text description. Some suggestions:

* More specifically label "+ sup". This is WT S. aureus sup? 

* In the text, more indicating which organism the genes are deleted from more often, would help the reader keep track. 

* In 3C, the blue control WT PA14 with StP in the absence of S. aureus supernatant? Does this result in a significant increase in P'11320 and suggest it is sufficient? 

* For figures where only one promoter is examined, it may help to indicated that on the Y-axis (P'11320 for 3C-F, I think). 

3. Model: Unless grown under swarming conditions, P. aeruginosa typically produce a single polar flagellum.

---

## [Decision Letter · Decision Letter 2]

6 May 2022

Dear Dr Khare,

Thank you for your patience while we considered your revised manuscript "Systematic identification of molecular mediators of interspecies sensing in a two-species bacterial community" for publication as a Research Article at PLOS Biology. This revised version of your manuscript has been evaluated by the PLOS Biology editors, the Academic Editor and one of the original reviewers.

Based on our Academic Editor's assessment of your revision, we are likely to accept this manuscript for publication, provided you satisfactorily address the remaining points raised and the following data and other policy-related requests.

IMPORTANT:

a) Please change your title to "Systematic identification of molecular mediators of interspecies sensing in communities of two frequently co-infecting bacterial pathogens"

b) You will see that reviewer #1 continues to raise some concerns related to those mentioned in the first round. The Academic Editor kindly discussed these issues with the reviewer, and has given me specific instructions as to how you should address these: "I would ask the authors to address the first two points that reviewer #1 raises, namely 1) to explain the rationale for selection of the clinical isolates that they now tested; and 2) to better discuss the variability of responses or their induction among clinical isolates, since not all of them behave the same way. As of now, their statement in the Abstract 'Cystic fibrosis clinical isolates of both S. aureus and P. aeruginosa also showed induction or responses, respectively, which suggests that these interactions are widespread among pathogenic strains' does not reflect this variability among isolates." We do not require you to address the remaining comments, or to do any additional experimental work.

c) Please address my Data Policy requests below; specifically, we need you to supply the numerical values underlying Figs 1CDE, 2AB, 3BCDEFG, 4ABCDE, 5ABDE, 6ABDE, 7ABCD, 8ABCD, S1, S2ABC, S3ABCD, S4ABC, S5AB, S6, S7, S8ABCD. Please supply these values, either as a supplementary data file or in a recommended repository. I see that you already present a large amount of data in your “Supplemental_File_2,” but it’s currently unclear how this relates to the Figs. Please also cite the location of the data clearly in each relevant main and supplementary Fig legend, e.g. “The data underlying this Figure can be found in S1 Data.”

We expect to receive your revised manuscript within two weeks. 

*Published Peer Review History*

*Press*

Sincerely,

Roli Roberts

Senior Editor,

rroberts@plos.org,

PLOS Biology

DATA POLICY:

Regardless of the method selected, please ensure that you provide the individual numerical values that underlie the summary data displayed in the following figure panels as they are essential for readers to assess your analysis and to reproduce it: Figs 1CDE, 2AB, 3BCDEFG, 4ABCDE, 5ABDE, 6ABDE, 7ABCD, 8ABCD, S1, S2ABC, S3ABCD, S4ABC, S5AB, S6, S7, S8ABCD. NOTE: the numerical data provided should include all replicates AND the way in which the plotted mean and errors were derived (it should not present only the mean/average values).

We require the original, uncropped and minimally adjusted images supporting all blot and gel results reported in an article's figures or Supporting Information files. We will require these files before a manuscript can be accepted so please prepare and upload them now. Please carefully read our guidelines for how to prepare and upload this data: https://journals.plos.org/plosbiology/s/figures#loc-blot-and-gel-reporting-requirements

DATA NOT SHOWN?

REVIEWER'S COMMENTS:

Reviewer #1:

This is a revised version of the manuscript by Zarrela and Khare. The results that are shown in this revised version are quite puzzling for this referee and I do not have a reasonable explanation for them. My comments below:

The strain selection: The authors included in this revised manuscript a selection of four clinical isolates of S. aureus and P. aeruginosa. These clinical isolates need to be introduced; it is important to know the physiological characteristics of the isolates, and where and how they were isolated. In their current state, they just have assigned a CF number with no other specification, which does not allow the reader to understand whether there is any important difference between these isolates (Table S14). Why they were selected among all from the CF foundation Isolate Core, where they were isolated, and whether there is any clinical data linked to these isolates (The CF foundation Isolate Core provides this information to researchers). This information is important if anyone from the scientific community wants to have access to these isolates.

The clinical strains show different results from the laboratory isolates. i.e. not all the clinical isolates induce all the reporters that the laboratory strain does. This applies to the S. aureus extracts (Fig. 7) as well as the P. aeruginosa isolates (Fig S6). This is something expectable and, as was discussed in our previous interaction, the response to CF isolates usually differs from laboratory strains and this is something that needs to be well considered. I disagree with the final conclusion of the authors that the response is commonly observed in laboratory and clinical isolates. There are differences, as is expected, and it is difficult to determine whether these differences are physiologically relevant, as it usually happens in these type of comparative studies. 

The methodology: The authors included in this revised manuscript new experiments showing that the response they described in this work is not related to the classical stress/starvation response that may result from supplementing the growth media with 25% of spent media. The authors showed that the supernatants collected from stationary S. aureus cultures can be added up to 25% to Pseudomonas cultures with no alteration of pH of growth rate. In the opinion of this referee, these results are quite shocking. As S. aureus cultures entered the stationary phase, nutrients are scarce and the presence of acid metabolism subproducts, such as citrate, acetate, or lactate reduce the pH of the medium and inhibit bacterial growth. The spent media from these cultures has unequivocally these features; nutrient deprivation and the accumulation of other stressors; low pH but also antibiotics (e.g. epidermin) and PSM that will affect pseudomonas growth and will trigger a stress response. None of this seemed to affect Pseudomonas growth. In fact, Pseudomonas cultures grown in fresh medium or in medium supplemented with 25% spent medium from staphylococcal cultures are well coordinated; both cultures started growing at the same time, showing a similar lag phase (one would expect, at least, a more extended lag phase in the culture supplemented with the spent media, as a consequence of the low nutrient and the presence of stressors) but also a similar growth rate. It is puzzling that both cultures keep a similar growth rate while showing important differences in nutrient availability (the growth in fresh medium showed a classical diauxic growth curve whereas the growth in spent media did not).

The authors show that the "addition of 25% (v/v) of just the salts from the medium did not induce any of the promoter-reporters (new Fig. S2C), indicating that nutrient deprivation did not underlie the responses we were seeing". It is really hard for this referee to understand how nutrient-deprivation conditions did not cause a response in Zn-deprivation, Fe-deprivation, TCA uptake, and acetoin catabolism, which are the four responses that are monitored in this work and are also responses strongly related to nutrient deprivation.

P. aeruginosa growth: The authors stated in their response letter that, according to figure S2B, supplementing growth medium with 25% spent media from S. aureus cultures did not affect P. aeruginosa growth because the culture medium used in this work was M63, which is a rich defined medium containing glucose and amino acids as carbon sources. To the understanding of this referee, the M63 medium is not a rich medium but a defined medium for gram-negative bacteria. Moreover, the preferred carbon sources of P. aeruginosa are TCA intermediates and amino acids; glucose is not an important carbon source for P. aeruginosa, unlike glycerol and acetate metabolites. Adding glucose as a carbon source makes the nutritional component of this medium poorer than the original formula containing glycerol.

---

## [Editor Report · Decision Letter 3]

17 May 2022

Dear Dr Khare,

On behalf of my colleagues and the Academic Editor, Victor Sourjik, I'm pleased to say that we can in principle accept your Research Article "Systematic identification of molecular mediators of interspecies sensing in a community of two frequently co-infecting bacterial pathogens" for publication in PLOS Biology, provided you address any remaining formatting and reporting issues. These will be detailed in an email that will follow this letter and that you will usually receive within 2-3 business days, during which time no action is required from you. Please note that we will not be able to formally accept your manuscript and schedule it for publication until you have completed any requested changes.

Sincerely, 

Roli Roberts

Roland G Roberts, PhD 

Senior Editor 

PLOS Biology

rroberts@plos.org